# An in vivo and in vitro spatiotemporal profile of human midbrain development

Dimitri Budinger [1,8], Pau Puigdevall [2,3,8], George T. Hall [3,8], Charlotte Roth [1], Theodoros Xenakis [3], Elena Marrosu[1], Julie Jerber[4], Alessandro Di Domenico[1], Francesca Picco[1], Helena Kilpinen [2,5,6], Sergi Castellano[3], Manju A. Kurian [1,7,9] ✉ & Serena Barral [1,9] ✉

The dopaminergic system has key roles in human physiology and is implicated in a broad range of neurological and neuropsychiatric conditions that are increasingly investigated using induced pluripotent stem cell-derived midbrain models. To determine similarities of such models to human systems, here we undertake single-cell and spatial profiling of first and second trimester fetal midbrain and compare it to in vitro midbrain models. Histological examination reveals that, by the second trimester, fetal midbrain tissue exhibits structural complexity comparable to that of adults. At the molecular level, single-cell profiling uncovers differences in cellular composition across models, with brain organoids most closely resembling late first trimester tissue — an observation supported by meta-integration of existing midbrain datasets. By reconstructing developmental trajectories of neuronal and astrocytic lineages, we map gene expression dynamics associated with maturation. Importantly, integration of spatial transcriptomics provides critical context for aligning organoid models, revealing that their spatial organization and intercellular signaling resemble the architecture and microenvironment of the second trimester midbrain. Ultimately, we leverage our findings to study Dopamine Transporter Deficiency Syndrome progression in patient-derived midbrain organoids, validating their relevance. Understanding the extent of human tissue recapitulation in midbrain laboratory models is essential to justify their use as biological proxies.

The midbrain is a complex brain region comprising nuclei, nerves and tracts involved in vision, sensory processing, cognitive function and motor control[1]. In this composite system, dopaminergic nuclei, located ventrally to the cerebral aqueduct, play an important role in reward, cognition and control of voluntary movement, throughout the ventral tegmental area (VTA) and substantia nigra[2,3]. Dysfunction of the midbrain dopaminergic (mDA) system has been associated with a wide spectrum of neurological and neuropsychiatric conditions, including

[1]Developmental Neurosciences, Zayed Centre for Research, UCL Great Ormond Street Institute of Child Health, University College London, London, United Kingdom. [2]Helsinki Institute of Life Science (HiLIFE), University of Helsinki, Helsinki, Finland. [3]Genetics and Genomics Medicine, UCL Great Ormond Street Institute of Child Health, University College London, London, United Kingdom. [4]Wellcome Sanger Institute, Wellcome Genome Campus, Hinxton, United Kingdom. [5]Faculty of Biological and Environmental Sciences, University of Helsinki, Helsinki, Finland. [6]Faculty of Medicine, University of Helsinki, Helsinki, Finland. [7]Department of Neurology, Great Ormond Street Hospital, London, United Kingdom. [8]These authors contributed equally: Dimitri Budinger, Pau Puigdevall, George T. Hall. [9]These authors jointly supervised this work: Manju A. Kurian, Serena Barral ✉e-mail: manju.kurian@ucl.ac.uk; s.barral@ucl.ac.uk

Parkinson's disease (PD), attention deficit hyperactivity disorder, autism, drug addiction, and childhood neurological diseases[4,5].

Despite conservation of several midbrain characteristics between species, the human midbrain has unique dopaminergic populations and developmental characteristics[6–9]. Recent advancements in single-cell RNA sequencing (scRNA-seq) and spatial transcriptomics have enabled the generation of comprehensive anatomical atlases for both the developing and adult human brain[10–14]. These human atlases have offered a detailed map of the cellular diversity and molecular properties in both fetal and adult midbrain. Despite these advances, a significant gap in knowledge remains, particularly regarding cellular diversity, spatiotemporal distribution, and the developmental trajectory in late embryonic to postnatal midbrain development.

In parallel, the advent of in vitro 2D models derived from induced pluripotent stem cells (iPSCs) has provided valuable insights into the human midbrain[15]. These models are also relevant for studying neurological disorders associated with midbrain dysfunction[16,17]. However, they do not capture the complete spectrum of cellular interactions and spatial organization that is characteristic of the human midbrain. In contrast, 3D midbrain-like organoids (MLO) more faithfully recapitulate the complexity of the human midbrain, where development of multiple cell types and structural organization more closely resembles the tissue organization observed during embryonic development[18–20]. Nevertheless, fundamental questions remain unanswered: at what stage do these iPSC-derived models deviate from the natural course of development, and if there is deviation, how effectively can they model neurodevelopmental and late-onset neurological disorders?

To explore the cellular diversity and developmental trajectory of the human midbrain, we employed a multimodal approach to profile midbrain tissue from human fetal samples at 6–22 post-conceptional weeks (PCW), using immunohistochemistry, scRNA-seq and spatial transcriptomics. To provide a comprehensive overview of the cellular landscape during this developmental window, we compared our findings to previously profiled human fetal and postmortem adult samples. Furthermore, we explored how this aligned with 2D and 3D iPSC-derived midbrain models, offering a valuable perspective on the relevance and fidelity of these in vitro systems to study human fetal midbrain development and model neurological disorders.

## Results

### Anatomical changes of the developing human ventral midbrain across the first and second gestational trimester

To investigate the spatiotemporal organization of the ventral midbrain during the first half of fetal human development, we conducted immunohistochemistry analysis on midbrain tissues from 6 to 22 PCW (n = 5 donors) (Supp Data S1), focusing our analysis on dopaminergic neurons. All images are available at high resolution in the MRC-Wellcome Trust Human Developmental Biology Resource (HDBR) Atlas (https://hdbratlas.org/gene-expression/midbrain.html).

The first gestational trimester (up to 12 PCW) represents the neurogenesis stage of midbrain dopaminergic (mDA) neurons[7,9,21–23]. Following regional specification of the neural tube, mDA progenitors express the transcription factors FOXA2[24] and LMX1A[25], both required for mDA neuron development (Supp Fig. 1A, B). At 6 PCW, proliferating mDA progenitors in the ventricular zone (VZ) express SOX2, Ki67 and OTX2 (Supp Fig. 1C–E)[9], while postmitotic mDA neurons express tyrosine hydroxylase (TH) and the microtubule-associated protein 2 (MAP2), in neurons located at the intermediate zone (IZ)[6,26] (Supp Fig. 1F). To further understand the spatiotemporal emergence of the SN and VTA, we investigated changes in the human midbrain during the second gestational trimester. We analyzed midbrain tissue from 12, 16, 19 and 22 PCW and detected a progressive increase in morphological complexity associated with the mDA neurons distribution over time (Fig. 1A; Supp Figs. 2–4). At 12 PCW the human midbrain is characterized by expansion of the ventral area (mantle zone) (Fig. 1B;

Supp Fig. 2A–E). At 16 PCW, we could identify the primordium of the SN and VTA within the mantle zone (MZ) of the basal plate, where TH+ mDA neurons occur for the first time (Supp Fig. 2F–K). During late stages of the second trimester, we observed increased complexity of the ventral midbrain and mDA nuclei organization. Further morphological specification of the SN and VTA was evident at 19 and 22 PCW, with presence of the red nucleus in the rostral area of the midbrain at both ages. At 22 PCW, the ventral midbrain resembled the complex architecture of the adult tissue[27] (Fig. 1C, D; Supp Figs. 3 and 4). To further investigate the distribution of midbrain neuronal populations during development, we analyzed the expression of OTX2 and PITX3, which are involved in maturation and specification of mDA neurons, as well as Calbindin (CALB1) and aldehyde dehydrogenase 1 Family Member A1 enzyme (ALDH1A1), which are expressed in the VTA and SN respectively[3,6,26,28]. OTX2 was mostly expressed across the first trimester, while we did not observe clear OTX2 expression at 12, 16 and 19 PCW (Supp Fig. 1E). PITX3 expression was observed in the ventral area of the developing midbrain across the earlier second trimester phases (12 and 16 PCW) (Supp Fig. 2). Spatial differential distribution of CALB1 and ALDH1A1/TH neurons was detected from 12 PCW to 22 PCW (Supp Figs. 2–4). During the early second trimester, sparse CALB1 mDA positive neurons were observed, whereas ALDH1A1 mDA neurons were already present in defined nuclei across the lateral ventral midbrain. At 16 PCW, TH+ neurons in the emerging SN were also expressing the G-protein-regulated inward-rectifier potassium channel 2 (GIRK2) (Supp Fig. 2J), pivotal for synaptic maturation in mDA neurons of the VTA and SN[29]. During 19 and 22 PCW, we observed an increase in complexity of both CALB1 and ALDH1A1 positive regions, with differential distribution across the rostral caudal axes at 22 PCW (Supp Figs. 3C–D and 4C–D). Interestingly, we did not observe OTX2 expression in the emerging VTA[30], but OTX2 positive nuclei were detected in the red nucleus at 22 PCW (Supp Fig. 4E). From 15 to 22 PCW, we observed synaptophysin (SYP) expression in the TH+ projections located in the SN and VTA[3] (Supp Fig. 5A). We also observed the presence of mDA neurons expressing the neurotransmitter GABA from 19 PCW, in line with recent transcriptomic observations (Supp Fig. 5A)[12].

Throughout the second trimester of gestation, we still observed mDA neurons located along the ventral midline migrating radially and not yet in their final location (Fig. 1B, C; Supp Fig. 3B)[26,28,31–33]. We also detected GFAP+ cells exhibiting an elongated morphology and oriented in a dorsal-ventral direction, suggesting the presence of cells with radial glial or astrocytic identity. At 22 PCW, GFAP+ cells in the ventral midline lost their oriented morphology (Fig. 1D). Both at 19 and 22 PCW, we observed mDA neurons undergoing tangential migration in the SN and VTA[26,28,31] (Fig. 1C, D). At 22 PCW, the tangential migration of mDA neurons was associated with GFAP+ cells. Radial and tangential migration were dependent on the maturation stage of mDA neurons. While at 19 PCW radial migrating mDA neurons did not express NEUN, a marker for neuronal maturity, it was expressed in radial migrating mDA neurons at 22 PCW, indicating increased neuronal maturity of these cells (Supp Fig. 5B). In contrast, tangentially migrating mDA neurons expressed NEUN at both 19 and 22 PCW (Supp Fig. 5C). Together, these observations indicate that across the first and second trimesters of gestation, the human midbrain undergoes major morphological changes, with increased complexity and neuronal maturity between 19 and 22 PCW, at which stage the fetal midbrain closely resembles the adult tissue[34,35].

### Cellular and morphological complexity of iPSCs-derived midbrain models

We next investigated the alignment of in vitro human midbrain models to fetal development. 2D mDA neuronal cultures[36] and 3D midbrain-like organoids (MLO)[19,37] were derived from two previously characterized human iPSC lines[36,38] (Fig. 2A). At 20 days of differentiation, we

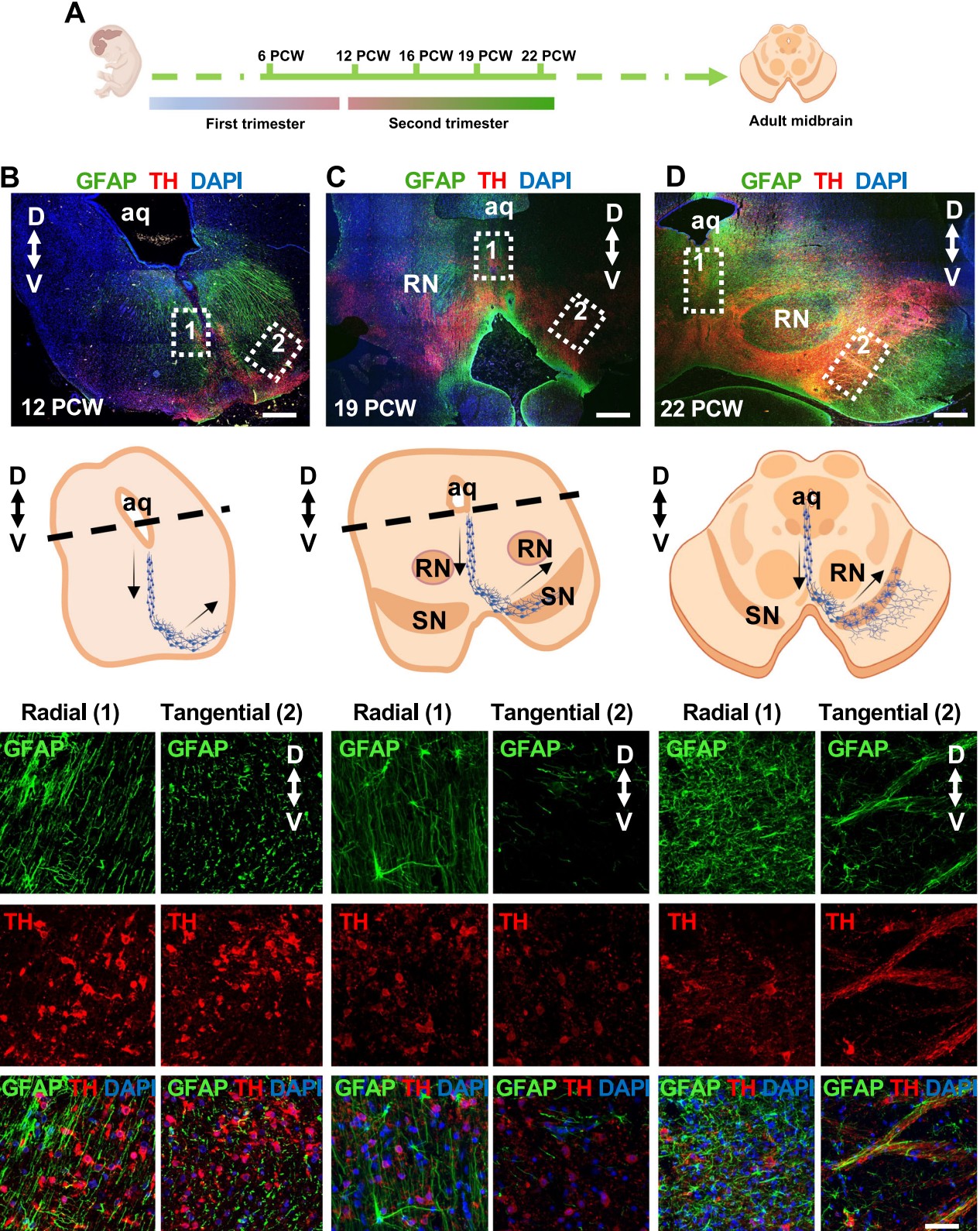

**Fig. 1 | Morphological changes of the ventral human midbrain during the second trimester. A** Experimental design indicating age of fetal samples analysed. *Created in BioRender. BUDINGER, D. (2025)* https://BioRender.com/f8y0e0d. **B**–**D** Immunofluorescence analysis of fetal midbrain tissue coronal sections collected at 12 (**B**), 19 (**C**) and 22 (**D**) PCW for TH and GFAP (*upper panel*). Nuclei are stained with DAPI. Scale bars *upper panel* = 300 μm (**B**) and 500 μm (**C**, **D**). Graphical representation of the ventral midbrain coronal section and TH positive cells radial and tangential migration (*middle panel*). *Created in BioRender. BUDINGER, D. (2025)* https://BioRender.com/yci1tqt. Higher magnification of radial and tangential migrating TH+ and GFAP+ neural cells at 12 (**B**), 19 (**C**) and 22 (**D**) PCW (*lower panel*). Dotted lines demarcate zoomed radial (1) and tangential (2) neuronal migration regions. Cerebral aqueduct (aq), dorsal (D), ventral (V), red nucleus (RN), substantia nigra[2]. Scale bars lower panel= 50 μm.

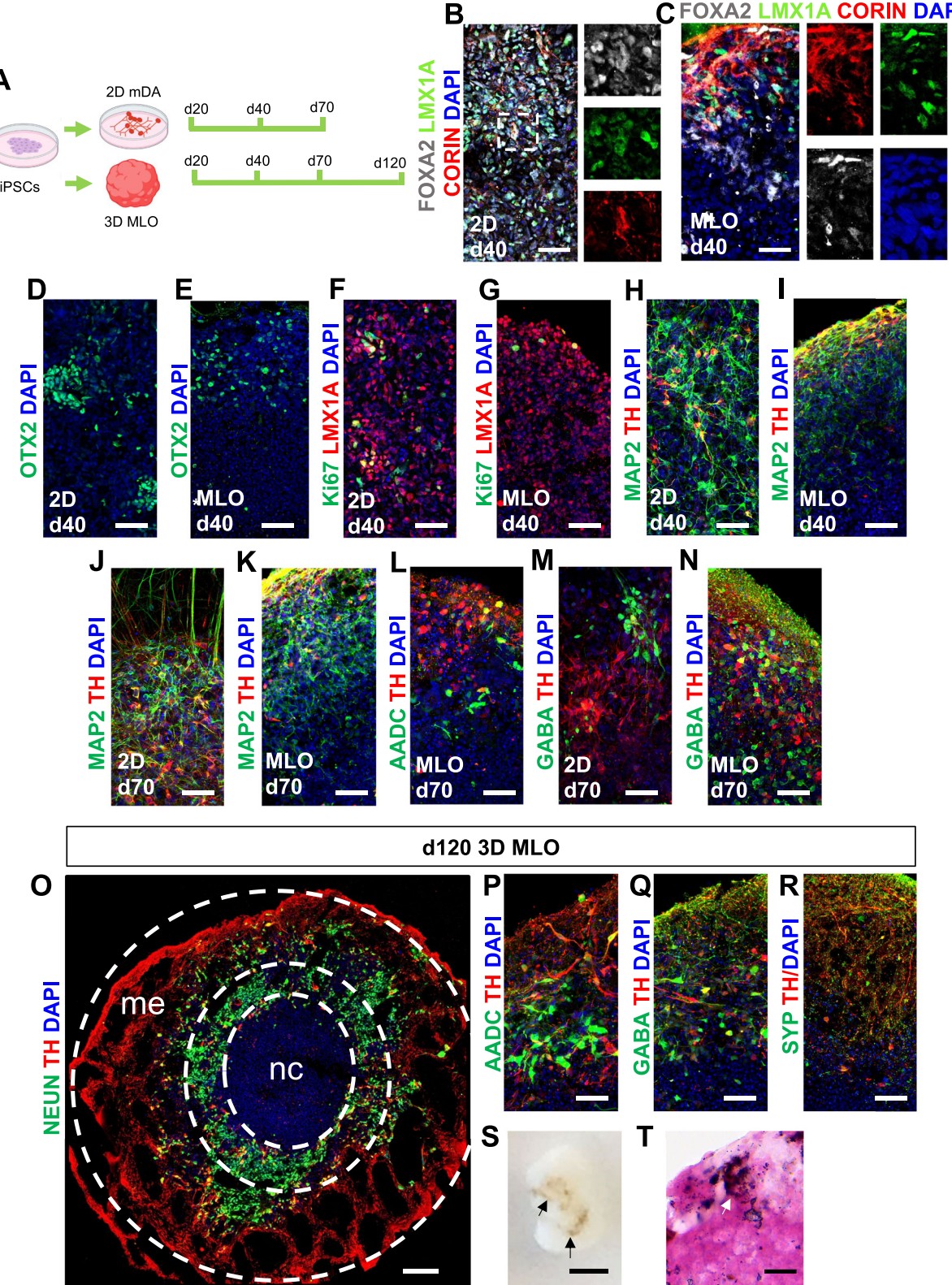

**Fig. 2 | In vitro derived 2D mDA neurons and 3D MLO models cellular composition. A** Experimental design for the in vitro system analysis. *Created in BioRender. BUDINGER, D. (2025)* https://BioRender.com/culevfq. **B–I** Representative images of 2D mDA neuronal culture and MLO at 40 days of differentiation. Cultures were stained for the midbrain-related and neuronal proteins FOXA2, LMX1A, CORIN, OTX2, TH, MAP2, and proliferative marker Ki67. **J–N** Representative images of 2D mDA neuronal culture and MLOs at 70 days of differentiation showing expression of TH, MAP2, AADC and GABA. **O–R** Immunofluorescence analysis of MLOs at 120 days of differentiations. MLOs were stained for TH, NEUN, AADC, GABA and synaptophysin (SYP). Nuclei are staining with DAPI. Scale bars = 100 μm (*left panel*) and 50 μm (*right panels*). **S** Bright field image visualizing melanin aggregates. Scale bar = 2 mm. **T** Fontana-Masson staining and haematoxylin and eosin (H&E) stain of MLO at 120 days of differentiation showing melanin precipitation. Scale bar = 50 μm. Necrotic core (nc), Matrigel embedding (me).

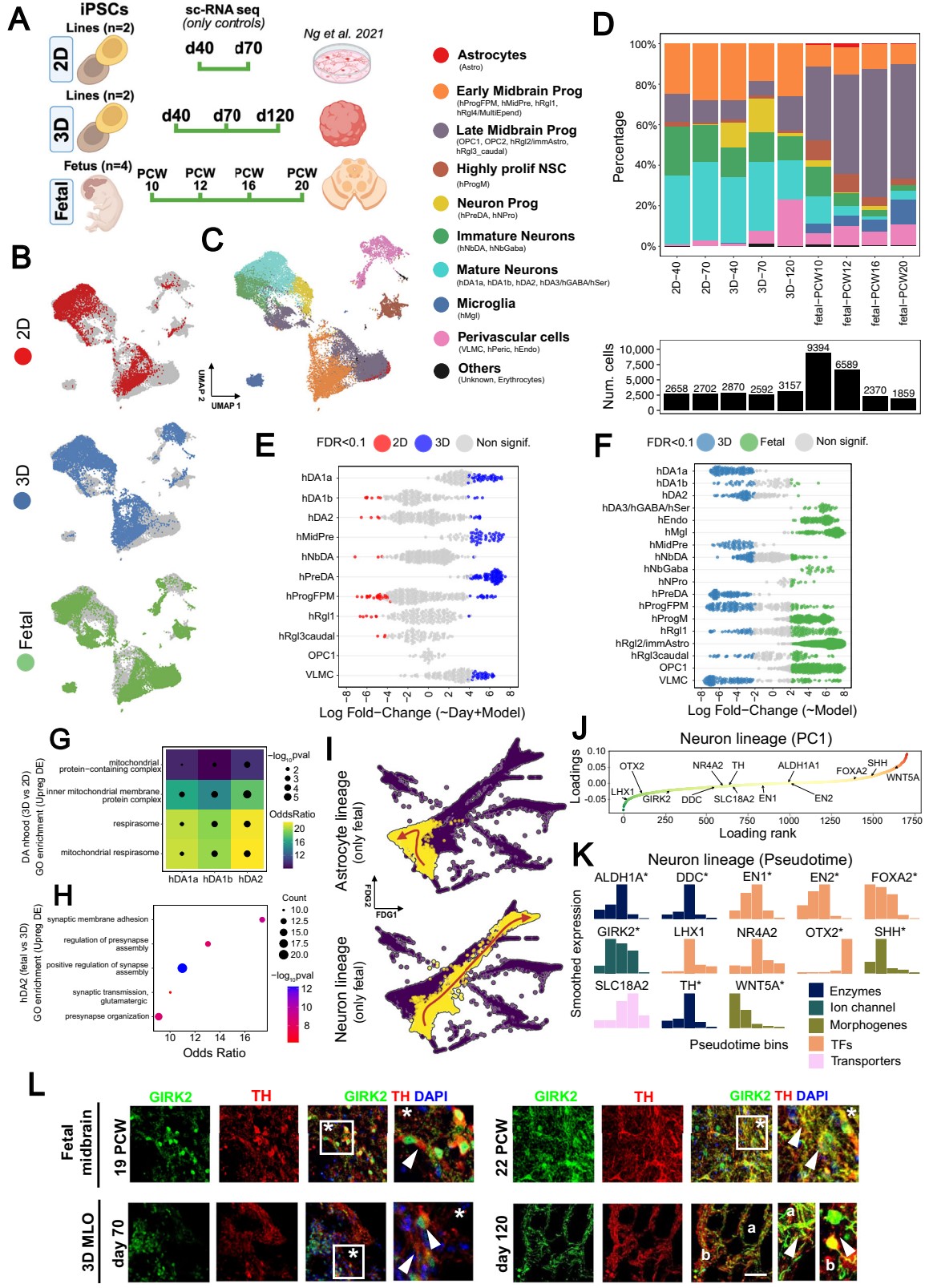

confirmed efficient neuralization and ventral midbrain identity, with mDA progenitors expressing both FOXA2 and LMX1A. MLOs had an expected central necrotic core where no FOXA2/LMX1A positive cells were detected[39,40]. LMX1A+ cells also expressed CORIN, SOX2 and the cell cycle marker Ki67, further confirming progenitor midbrain identity. We also observed MAP2+/TH+ cells in both 2D and 3D culture, suggesting that a fraction of mDA neurons have already reached post-

mitotic maturation at that stage (Supp Figs. 6 and 7). At 40 days of differentiation, FOXA2, LMX1A and OTX2 expression was maintained in mDA progenitors in both 2D and 3D models (Fig. 2B–E; Supp Figs. 6I, K and 7I, K), while we observed fewer Ki67+ cells than at day 20 (Fig. 2F, G; Supp Figs. 6B, F, J, 7B, F, J and 8A). MAP2+/TH+ cells were present in both 2D and MLOs (Fig. 2H, I; Supp Figs. 6L and 7L). At 70 days, co-expression of MAP2, AADC, TH and NEUN, indicated the

**Fig. 3 | Single-cell RNA-seq profiling of in vitro and in vivo fetal samples.**
**A** Scheme of the in vitro (2D, 3D) and fetal samples profiled with droplet-based scRNA-seq. Only control samples were considered for all models: 2 iPSC lines from pediatric controls were differentiated to a midbrain dopaminergic fate, either as a 2D culture (days 40, 70) or 3D organoids (days 40, 70, 120). As for fetal samples, the midbrain sections of four donors were profiled at single timepoints: 10 and 12 PCW for the first trimester, and 16 and 20 PCW for the second trimester. *Created in BioRender. BUDINGER, D. (2025)* https://BioRender.com/pt104zy. **B** UMAP visualization based on gene expression colored by in-vitro (2D and 3D) or in-vivo dopaminergic models (fetal tissue). **C** UMAP visualization (as in b) with cells grouped in 10 broad categories (low-resolution annotation), encompassing the 24 cell types defined by the high-resolution annotation. The largest cell clusters correspond either to neuronal cell population (top-left) or to radial glia/floor-plate progenitors (bottom-right). **D** Upper: cell type composition per model and time point. Lower: number of cells profiled in each corresponding group to estimate cell type composition. **E, F** Differential abundance (DA) analysis comparing cell types between 3D vs 2D in-vitro models (**E**), and between fetal samples and the 3D model (**F**). Each point represents a neighborhood (group of cells connected by an edge in the KNN graph), with coloured points indicating significant DA (SpatialFDR < 0.1). Only in (**E**) the sampling timepoint is included as a covariate. **G** The mitochondrial respirasome is commonly upregulated in 3D versus 2D models across neuronal populations (hDA1a, hDA1b and hDA2) indicating a prominent role of mitochondria metabolism

in the maturation of dopaminergic neurons. Differential gene expression between 3D and 2D cells was calculated within the same cell type (neighborhood group), accounting for timepoints as a covariate (adj P.val<0.05, FC > 1.75, hypergeometric test for over-representation analysis implemented in gprofiler2 (gost)). **H** Fetal hDA2 cells show an upregulation of synaptic-related pathways when compared with the 3D hDA2 cells (only the top-5 biological processes of the gene ontology enrichment are shown, pval<0.05, minSize=25, hypergeometric test for over-representation analysis implemented in the *gprofiler2* R package). **I** Force-directed graph (FDG) representation of inferred trajectories for astrocytes (top) and neurons (bottom). Cells highlighted in yellow belong to the inferred trajectory, and the arrow indicates the direction of increasing pseudotime. All cells shown in **B**, **C** and **I** were profiled based on experimental design available in Supp Fig. 10A. **J** PC1 loadings of highly variable genes defined along the neuronal lineage. Genes whose expression correlates with neuronal maturation or a progenitor-like state are shown in green and red, respectively. **K** Smoothed expression profiles of a curated set of neuronal marker genes (same as in **J**), across five evenly spaced pseudotime bins. Asterisks indicate significant start-to-end gene expression changes. **L** Immunofluorescence analysis of fetal midbrain at 19 and 22 PCW and MLOs at 70 and 120 days of differentiation for TH and the potassium channel GIRK2. Arrowheads indicate colocalization of GIRK2 and TH either in the soma or in the neurites. Nuclei are stained with DAPI. Scale bar= 50 μm.

presence of post-mitotic mature mDA neurons (Fig. 2J–L; Supp Figs. 6O and 7Q–T, 7W). We also observed the presence of neurons expressing the GABAergic neuronal marker GABA, and cells expressing GFAP (Fig. 2M, N; Supp Figs. 6P, R and 7U,V). 2D mDA neurons were maintained for a maximum of 70 days, whereas MLO differentiation allowed for an extended culture of up to 120 days. At this later stage of MLO differentiation, we observed the presence of NEUN+ and TH+ mDA neurons, which projected neurites outwards of the organoids (Fig. 2O). We also observed expression of mature midbrain markers including AADC, GABA and SYP (Fig. 2P–R; Supp Fig. 8B–K). Gene expression of *LMX1B, EN1, EN2, OTX2, NURR1, TH, PITX3, DAT* and *SNCA* also confirmed midbrain progenitor identity and mDA maturation over time and across both 2D mDA neurons and MLO cultures (Supp Fig. 8L). Fontana-Masson staining revealed the presence of intracellular and extracellular neuromelanin, visible as a dark granular pigmentation, suggesting the specification and mature profile of mDA neurons of the SN pars compacta in these organoids (Fig. 2S, T). Thus, our iPSC-derived mDA neuronal 2D and 3D systems recapitulate the major developmental steps of human midbrain development and can be harnessed to model these processes.

## Single cell profiling of the developing human midbrain

To further understand both the cellular and molecular complexity of the developing human midbrain, we performed droplet-based scRNA-seq of the human fetal midbrain covering first-trimester (10 and 12 PCW) and second-trimester gestational ages (16 and 20 PCW), jointly with iPSC-derived 2D mDA neurons (days 40 and 70) and MLOs (days 40, 70, and 120). Overall, we profiled a total of 34,191 cells from six donors (1,346 − 9,394 cells per donor, summed across 2 to 9 replicates) that were pooled in 13 samples, each corresponding to a 10X library prepared from a single channel (Fig. 3A, B; Supp Fig. 10A; Supp Data S2). Donor identity per cell was deconvoluted using demuxlet[41], samples were merged and normalized with Seurat[42] and integrated with Harmony[43]. We annotated cells using a two-tiered approach based on high and low-clustering resolution (Supp Data S3), identifying 10 major groups and 24 cell types according to established markers[6,18] (Fig. 3C, D; Supp Figs. 9A, B and 10B). Three of the cell types corresponded to either perivascular identities, including endothelial (hEndo), pericytes (hPeric)[6], and vascular leptomeningeal cells (VLMC)[18]; or to glial cell types, including microglia (hMgl), astrocytes (Astro) and two oligodendrocyte progenitor cells (OPC1 and OPC2). We distinguished OPC1 and OPC2 populations by the higher

expression of *SOX10* in the OPC1 cluster (Supp Fig. 9A). In addition to the two OPC populations, we identified two other late midbrain progenitors cell types: hRgl2/immAstro and hRgl3caudal. Early midbrain progenitors were composed of four radial-glia like clusters: hRgl1 showing midbrain floor plate identity, expressing *FOXA2, CORIN, SOX6* and *PBX1*; hRgl4/multiEpend with a *CCNO* upregulation[44] linked to cilium organization/assembly and axoneme assembly (ciliary core); progenitor medial floorplate (hProgFPM), expressing *FOXA2; and* midbrain precursors (hMidPre). We further identified a highly proliferative neural stem cell-like identity, annotated as progenitor midline (hProgM), and two neuronal progenitors, the dopaminergic precursors (hPreDA) and the neuronal progenitors (hNPro). Two populations of immature neurons were identified, either with dopaminergic commitment (hNbDA, expressing *TH, LMO3, DDC, CALB1* and *PBX1);* or with GABergic commitment (hNbGABA). Our clustering separated mature dopaminergic neurons into four subpopulations: hDA1a, hDA1b, hDA2 and hDA3/hGABA/hSer. hDA1a and hDA1b were annotated based on their relative expressions (low in hDA1a, high in hDA1b) of *TH, KCNJ6* and *DCX* genes, while hDA2 expressed *TH, KCNJ6, CALB1* and *DDC*. The hDA3/hGABA/hSer was a mixed cluster population, expressing increased levels of *LMO3, PBX1* and *KCNJ6*, jointly with high levels of *GAD1, GAD2* and *MEIS2* (GABAergic-related genes) and *GATA3* (involved in serotonergic neuron differentiation). Finally, we identified two other clusters of cells, one belonging to erythrocytes and another with an unknown origin, which did not align with any of the previous cell identities. This cluster was abundant in genes related to collagens (*COL2A1, COL9A3, COL9A1, COL9A2, COL11A1*) which may represent connective tissue cells[45].

After cell type annotation, we estimated cell type proportions for each model and time point, with the most accurate estimates coming from first-trimester fetal samples, which had at least twice as many cells (Fig. 3D). When comparing in vivo with in vitro models, we detected a larger proportion of cells with dopaminergic fate in vitro, either mature or immature neurons, due to the more directed differentiation towards the midbrain lineage. Conversely, microglia and late midbrain progenitors were more abundant in fetal samples. The underrepresented neuronal population in fetal samples might also reflect the increased complexity of the fetal tissue or technical limitations associated with scRNA-seq tissue processing. Since our cells followed a continuous trajectory rather than clustering discretely, we used the MiloR framework[46] to test for differential abundance of cell types between the three models (Fig. 3E–G).

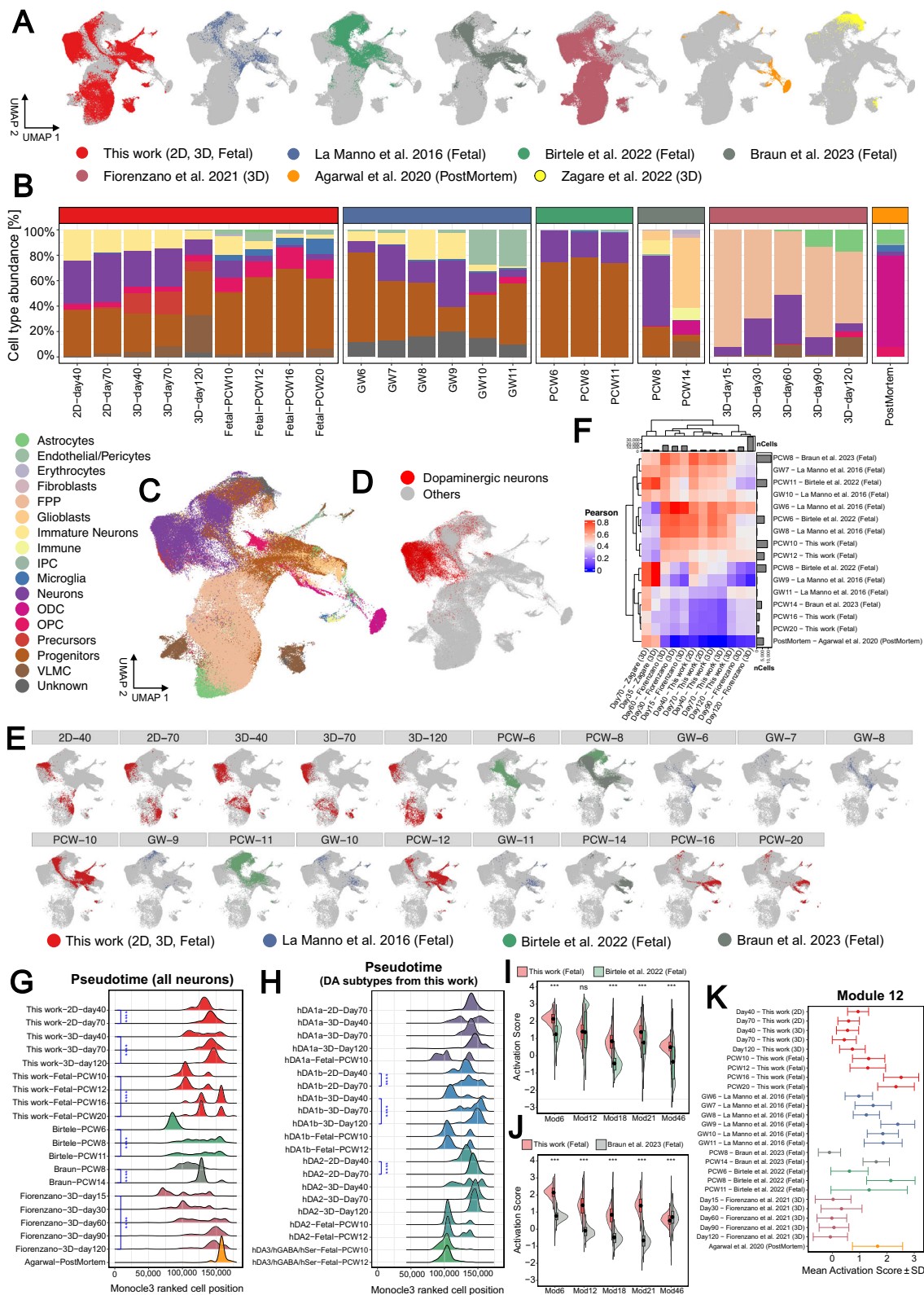

By comparing 3D models against 2D ones, and considering the differentiation day as a covariate, we observed an enrichment of the dopaminergic population hDA1a and some of the precursors (hMidPre, hPreDA) in MLO (Fig. 3E). On the other hand, some of the other dopaminergic neurons (hDA1b, hDA2) had subsets enriched in both models, indicating substantial cell state differences. Despite this, when running differential gene expression between 3D and 2D in the three

neuronal populations, they all converged in the upregulation of the mitochondrial respirasome (Pval<0.05, FC>1.75), suggesting a more prominent role of mitochondria metabolism in the dopaminergic neurons from MLO (Fig. 3G, Supp Data S4). We then compared fetal samples with 3D MLOs, observing a strong enrichment of the mixed hDA3 population and a depletion of the other neuronal subtypes during in vivo development (Fig. 3F). Differential gene expression

**Fig. 4 | Meta-integration of multiple dopaminergic scRNA-seq datasets and analysis of pseudotemporal dynamics. A** UMAP visualization based on the integrated gene expression (MNN correction) of our dataset with in-vitro 3D dopaminergic models (Fiorenzano et al. 2021, Zagare et al. 2022), in-vivo fetal data (La Manno et al. 2016, Birtele et al. 2022 and Braun et al. 2023), and adult post-mortem samples (Zagare et al. 2022). The final integrated dataset contains 194,521 cells. **B** Cell type composition per dataset and time point, expressed in percentage. For comparability between datasets, a unified cell type annotation (17 identities) has been defined (legend on 3rd row). Abbrev: floor plate progenitors (FPP), intermediate progenitor cells (IPC), oligodendrocytes (ODC), oligodendrocyte progenitor cells (OPC) and vascular leptomeningeal cells (VLMC). **C, D** UMAP visualization from (a) colored by either cell types (unified annotation) or just dopaminergic neurons, respectively. **E** UMAP visualization from (a) colored by dataset and timepoint. Only cells collected from this work and fetal samples are highlighted. **F** Transcriptomic similarity (Pearson's correlation of MNN corrected gene expression, n = 2000 features) between fetal gestational timepoints and in vitro 3D dopaminergic differentiation timepoints. Abbrev: gestational week (GW), post-conceptional week (PCW). **G** Ridge density plot of pseudotime distribution for neurons, colored per dataset and displayed per timepoint (each row represents a different timepoint). Differences in pseudotime distribution were statistically assessed across six within-dataset comparisons (*This work*: 2D-day 40 vs 2D-day 70; 3D-day40 vs 3D-day120; fetal-PCW10 vs fetal-PCW20; *Birtele*: fetal-PCW6 vs fetal-PCW11; *Braun*: fetal-PCW8 vs fetal-PCW14; and *Fiorenzano*: 3D-day15

vs 3D-day120). Significance was evaluated using a two-sided Wilcoxon test (all comparisons showed $p < 2.22 \cdot 10^{-16}$, indicated by ****). **H** Ridge density plot of the pseudotime distribution for dopaminergic neurons subtypes annotated in this study. The plot is colored by subtype and displayed per timepoint (each row represents a different timepoint), shown only when enough cells ( > 50) are available per combination. Differences in pseudotime distribution were statistically assessed across three comparisons (*hDA1b*: 2D-day40 vs 2d-day70; *hDA1b*: 3D-day40 vs 3d-day120; and *hDA2*: 2D-day40 vs 2D-day70). Significance was evaluated using a two-sided Wilcoxon test (all comparisons displayed $p < 2.22 \cdot 10^{-16}$, indicated by ****). **I, J** Split violin plots showing activation scores (aggregated gene expression) for the five neuron-specific pseudotime-dependent gene modules, comparing fetal neurons from this study with those from Birtele et al. 2022 (**I**) or from Braun et al. 2023 (**J**), respectively. All comparisons displayed $p < 10^{-3}$ (***), except for Module 12 in (**I**) with p = 0.418. Significance was assessed using a two-sided one-way permutation-test (10,000 label permutations per test, mean p-value for 100 resamplings reported). Violin plots display the full data distribution per group, with embedded boxplots showing the median (center line in black), the interquartile range (box, 25th–75th percentiles) and whiskers extending to 1.5x interquartile range. **K** Activation score distribution at single-cell level for the neuron-specific module 12, colored by dataset and displayed per timepoint (each row represents a different timepoint). The plot shows mean ± standard deviation across individual neurons (n = 33,504), as defined by the unified annotation.

analysis between fetal samples and MLOs revealed an upregulation of synaptic-related pathways in hDA2 fetal cells (synaptic membrane adhesion, presynaptic organization) (odds ratio, OR > 10) (Fig. 3H; Supp Fig. 10C, Supp Data S5). Conversely, hDA2 fetal cells showed downregulation of transcription and translation-related pathways, usually involving the Golgi vesicle-transport (Supp Fig. 10D). As previously observed, fetal samples exhibited a higher abundance of microglia, radial glia and oligodendrocytes progenitors, highlighting greater differences between fetal samples and 3D organoids than differences between the two in vitro models (3D vs 2D). We also tested for differential abundance between first and second-trimester gestational ages and found microglia enriched in late fetal, while immature neurons were enriched in early fetal stages (Supp Fig. 10E).

Profiling first and second trimester fetal tissue provided a unique opportunity to reconstruct developmental trajectories through pseudotime analysis. Using an approach based on a partition–based graph abstraction (PAGA)[47], we isolated the neuronal and astrocytic lineages anchoring them in early midbrain progenitors (Fig. 3I; Supp Fig. 11A–E, 11J, K).

Within these isolated lineages, the first principal component (PC1) captured the expected emergence of cell type identities along the trajectory (Supp Fig. 11F–H, L, M). We then examined the PC1 loadings of highly variable genes (HVG) to identify those most associated with either end of the differentiation axis. In the neuronal lineage, genes associated with neuronal development and synaptic function, such as *STMN1/2*, *SYT1* and *NSG2*, were more expressed in mature neurons, whereas *ID4*, *TTYH1* and *VIM* were more expressed in early midbrain progenitors (Supp Fig. 11I). Notably, transcription factors such as *LHX1* and *OTX2*, as well the ion channel *GIRK2*, were strongly correlated with mature neurons, while the *WNT5A* and *SHH* morphogens, were more specific to early midbrain progenitors (Fig. 3J). In the astrocyte lineage, genes involved in synaptic clearance (*SLC1A2*, *SLC6A11*) and neuroprotection (*CST3*, *MT3*) were among the highest PC1 loadings (Supp Fig. 11O). We also observed a significant correlation between the developmental age of the fetal samples with the inferred pseudotime (Supp Fig. 11N).

To further characterize gene expression dynamics along each lineage, we applied tradeSeq[48]. This approach revealed temporal expression profiles of key developmental genes, including morphogens, transcription factors, enzymes, transporters and structural components (Fig. 3K; Supp Fig. 11P). Importantly, canonical dopaminergic markers, such as *GIRK2* and *TH*, peaked at mid-trajectory, while

*OTX2* expression significantly increased later in differentiation (Supp Data S15). In astrocytes, the expression of many relevant transporters (*AQP4*, *GJA1*) peaked only at the end of the differentiation, being significantly increased in two solute carrier transporters (*SLC1A2*, *SLC6A11*) (Supp Data S16).

To further investigate the transcriptomic differences observed between fetal and in vitro systems, we used immunofluorescence to analyze the spatial expression of tyrosine hydroxylase (TH*)* and the GIRK2[29]. Comparing SN samples at 19 and 22 PCW, we observed GIRK2 primarily in the soma of TH+ neurons at 19 PCW, shifting to cytoplasmatic and dendritic localization at 22 PCW (Fig. 3L). In our 2D in vitro system, GIRK2 expression remained confined to the cellular soma, whereas in d120 MLOs, it localized to neurites (Supp Fig. 10F), closely resembling the expression pattern seen in the 22 PCW fetal sample. These findings indicate that, while there are differences in cell type composition and transcriptomic profiles between in vivo and in vitro systems, certain cellular phenotypes – such as GIRK2 localisation – suggest that these models can reach a comparable degree of maturity for specific features of midbrain development.

## Transcriptional dynamics of integrated atlases of developing human midbrain in vivo and in vitro

To comprehensively assess the strengths of our scRNA-seq results, we integrated our dataset with previously published ones covering fetal development[6,7,11], in vitro dopaminergic models[8,18] and adult post-mortem samples[49]. In total, we integrated 194,521 cells using the mutual nearest neighbours implementation (fastMNN) from Seurat v5. The integration showed alignment of cells upon their origin (2D, 3D MLO, fetal), except for the Zagare dataset[8] and, as expected, for the post-mortem adult brains[49] (Fig. 4A).

To compare cell type composition differences, we established a unified cell type annotation based on the published annotations of the datasets (Fig. 4B–D, Supp Data S6). Neuron abundance varied significantly between datasets, with more stable proportions in vitro (min-max: 34.1%–39% in 2D; 6.1%–38.9% in 3D) compared to fetal samples (min-max: 1.2%–55.3%). Generally, neuron proportions decreased at later stages of fetal development, with a simultaneous increase in glial cells, microglia and oligodendrocyte cell lineages. Unsurprisingly, the proportion of neurons in adult post-mortem sample was only 3.3%. Fetal datasets exhibited large variability in cell type composition which became more pronounced at later gestational stages, potentially due to a combination of both biological and

technical factors. For instance, apart from post-mortem samples[49], microglia was only detected in our study, while oligodendrocytes were only present in Braun et al. 2023 (PCW14). Still, the most abundant identity in the midbrain across all datasets and timepoints were progenitors.

We identified a developmental temporal axis among samples that correlates with the first component of the UMAP projection (Fig. 4E). In this sequence, in vitro models precede first-trimester fetal samples, followed by second-trimester samples. Thus, cells from the oldest sample (20 PCW) are positioned at the right end of this developmental axis. Based on this observation, we compared the transcriptional alignment of iPSC-derived models to fetal samples using pseudobulk batch-corrected values. Our findings reveal that iPSC-derived models begin to diverge from fetal development around 10 PCW, consistently in this and the Fiorenzano studies[18] (Fig. 4F). Based on transcriptional similarity, days 15–70 in MLO strongly correlated with gestational weeks (GW6-GW8) fetuses from La Manno[6] (average R = 0.68, Pearson's correlation), while days 90–120 in MLO correlated with 12 PCW fetus from this study (average R = 0.47). In contrast, one group has reported organoids (days 35–70) that exhibit similarities with adult post-mortem brains[8], a feature not shared by any of the other considered organoid models.

Given the presence of this potential developmental axis, we inferred a pseudotime trajectory on the integrated dataset using three methods: Slingshot, Destiny and Monocle3[50–52] (Supp Data S7). Among these, Destiny and Monocle3 showed a significant alignment on the final trajectory ($R^2$ = 0.737). Consequently, we chose the Monocle3 output for downstream analysis, anchoring the start of the trajectory at floor plate progenitors (Supp Fig. 12A, B). The trajectory followed the expected progression of cell type identities along the developmental axis, moving from radial glia, progenitors to precursors, intermediate progenitor cells, immature neurons, and neurons. The sampling timepoint also influenced the pseudotime inference, even within the same cell types (Supp Fig. 12C, D). For instance, neurons in Braun samples[11], were, on average, associated with a larger pseudotime at PCW14 compared to PCW8. Likewise, progenitors at PCW10 appeared older than those from d120 3D organoids. Interestingly, this progression was not maintained when comparing neurons from these two samples (Supp Fig. 12E, F). To analyze the neuronal developmental axis specifically, we compared the pseudotime distribution of neurons across different datasets and timepoints. In each dataset, older samples exhibited a skewed distribution toward larger predicted pseudotime values, both in differentiation and in development (Fig. 4G). This pattern was observed in 3D-organoids from day 120 and in late fetal samples (i.e. PCW20 in this study and PCW14 in Braun dataset). Among dopaminergic neuron subtypes from our study, the pseudotime values again showed some association with developmental time, especially for the hDA1b subtype in iPSC-derived models (Fig. 4H).

To further understand the transcriptional dynamics of the dopaminergic system, we identified 50 gene modules of co-regulated genes whose expression changed with pseudotime (Supp Data S8). We then computed activation scores for each cell to identify modules specific to cell type identities (Supp Fig. 13A). While not all modules were cell type-specific, some showed high activation in certain cell types, such as module 1 in ODC and module 38 in IPC. We focused on the five most highly activated modules for neurons (modules 6, 12, 18, 21 and 46), which were also activated in immature neurons and other cell types (Supp Fig. 13B). For instance, module 12 was highly expressed in progenitors, while module 18 was highly expressed in precursors. To identify the biological processes behind these neuron-enriched gene modules, we performed gene ontology enrichment analysis with *gProfiler2*[53]. We found significant enrichment in cellular components (p < 0.05) for three of the modules: synapse in module 6, voltage-gated calcium channel complex in module 12, and axon terminus in module 18 (Supp Fig. 13C).

The neuron-enriched modules showed activation differences across fetal cells from different publications (Fig. 4I, J; Supp Fig. 13D). In our study, all modules were significantly upregulated when compared to other datasets, except for module 12, which showed no difference with the Birtele study[7] and was downregulated when compared to La Manno samples[6] (*Fisher-Pitman permutation test*). This overall upregulation in our study can be partially attributed to the inclusion of second-trimester fetal samples. In line with this, module 12 is upregulated with developmental time, underscoring the relevance of calcium metabolism in neuron maturation (Fig. 4K). Conversely, module 46 is downregulated with developmental time, explaining the lower activation scores in our fetal cells compared to the Braun dataset[11] (Supp Fig. 13G). Comparing module activation scores within our dataset between in vitro models and fetal cells (Supp Fig. 13E), we observed higher activation in vitro in those modules whose scores anticorrelate with developmental time (modules 18 and 46), and lower activation for those that actually correlate (modules 6 and 12). Using the 3D organoid cells from Fiorenzano[18] as a control comparison, we observed higher activation of the five neuron-enriched modules in our in vitro models (Supp Fig. 13F), particularly in module 18, highlighting a more prominent role of axon terminus, necessary to release neurotransmitters. However, those differences are limited by the low number of neurons at late fetal stages resulting in more uncertain activation scores, as well using uncorrected (log-normalised) expression values to include all available genes. Overall, our results highlight consistent developmental trajectories of in vitro models across studies, indicating that their transcriptional profiles most closely resemble those of the first trimester of gestation.

## Three-dimensional in vitro midbrain models recapitulate the spatial cellular organisation of the second-trimester human fetal midbrain

While we have identified significant changes in the transcriptional dynamics of developing dopaminergic neurons at single cell level, spatial information is necessary to better understand human midbrain dopaminergic subtype development. To resolve the spatiotemporal developmental events of the ventral midbrain, we performed spatial transcriptomic analysis using the 10X Genomics Visium platform. We collected 7 PCW sagittal sections, along with 11 and 17 PCW coronal sections of the ventral human midbrain (n = 3 donors) (Fig. 5A, B; Supp Fig. 14; Supp Data S9). Additionally, we analyzed MLOs at 40, 70 and 120 days of differentiation in groups of six per capture area on the Visium slide (Fig. 5A, B; Supp Fig. 15).

We investigated the spatial distribution of cell types in the developing midbrain and MLOs by performing cell type deconvolution using our scRNA-seq dataset. We observed agreement between deconvoluted cell types and anatomical regions, with the scRNA-seq-defined cell types distributed along the dorso-ventral axis of the developing human midbrain (Fig. 5C; Supp Figs. 16 and 17A, B). Eight immature clusters were in proximity to the VZ and cerebral aqueduct, both at 7 and 11 PCW, respectively. We identified all hRgl clusters (1–4) and progenitor clusters (hProgFPM, hMidPre, hNPro, hProgM) in the VZ. Neuronal precursors (hPreDA, hNbDA, hNbGABA) localized in the IZ and mantle zone (MZ) of the midbrain at 7 PCW. At later stages of development (11 and 17 PCW) hPreDA were mostly localized at the ventral midline of the midbrain in proximity to the VZ. A more diffuse distribution was observed for hNbDA and hNbGABA, with the latter mostly localizing in the dorsal section of the developing midbrain at 11 PCW. hDA1a neurons were present in similar areas as hNbDA, though at distinct spots. hDA1b and hDA2 showed different spatiotemporal localizations during development. While at 7 PCW both populations were in the most ventral area of the midbrain where mDA neurons are localized, at 11 and 17 PCW a more scattered distribution was observed, with cells distributed at the ventral midline and the alar section of the tissue. In the organoids, hRgl1 cells were located

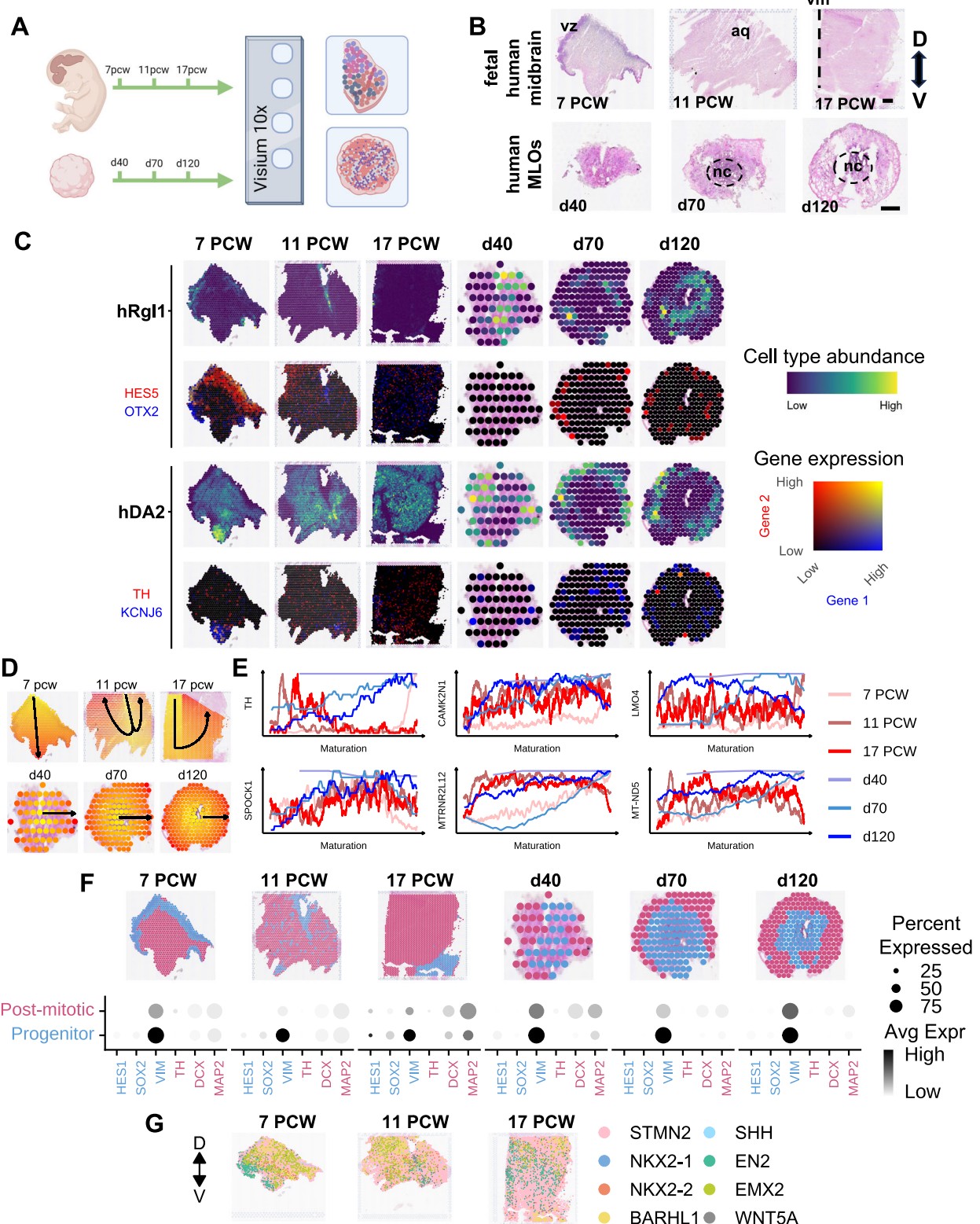

**Fig. 5 | Spatial distribution of midbrain cell types across development.**
**A** Experimental design for the spatial transcriptomics. *Created in BioRender. BUDINGER, D. (2025)* https://BioRender.com/f0bwfhn. **B** Haematoxylin and eosin (H&E) stain of human fetal samples at 7, 11 and 17 PCW and MLO samples at 40, 70 and 120 days of differentiation. Scale bars = 300 μm (*upper panel*); 500 μm (*lower panel*). Ventricular zone (VZ), cerebral aqueduct (aq), dorsal (D), ventral (V), necrotic core (nc). **C** Abundance of hRgl1 and hDA2 cells in tissues and organoids, as estimated by the Cell2Location deconvolution algorithm. Expression of two marker genes for each cell type are displayed below. **D** Manually annotated maturation paths on tissues and organoids. **E** Moving averages of the expression of genes that are differentially expressed with respect to maturation paths, calculated over a window of 50 spots along the path of maturation. **F** Progenitor and post-mitotic cell populations in tissues and organoids, defined by clusters annotated according to anatomical features and marker genes (displayed below). **G** Approximations of spatial locations of RNA molecules associated with midbrain patterning and specification.

towards the center while the more mature hDA2 were positioned peripherally.

Ontogeny of mDA neurons is defined by the spatial localization of the neurons in the tissue and follows specific migratory stages. To further investigate developmental cellular trajectories across the tissue and MLOs, we identified genes that are differentially expressed along the path of maturation (Fig. 5D). For each sample, we manually annotated the maturation path and then used it as a covariate in the "cell type-specific inference of differential expression" (C-SIDE) algorithm, which controls for the influence of different cell types at different spots (i.e. differentially expressed genes are identified for each cell type)[54]. We identified a total of 83 and 52 genes as differentially expressed along the paths of maturation in fetal midbrain tissue and organoids, respectively (Supp Data S10). For all samples we then plotted the moving average expression along the maturation path of genes reported to be involved in mDA neurons maturation (Fig. 5E). Similar trends in gene expression dynamics were mostly observed between 7 PCW fetal samples and day 70 MLOs (Fig. 5E). We detected increased expression along the maturation pathways of the dopaminergic neurons related genes *TH*, *CAMK2N1* and *LMO4*[6,8,18]. Similar expression dynamics were observed across all samples for the extracellular-matrix-related gene *SPOCK1*, coding for SPARC/osteo-nectin, cwcv and kazal-like domains proteoglycan 1 (SPOCK1) protein, which plays a role in brain development and has been indicated to been specifically involved in murine mDA neurons specifications[55,56]. The mitochondrial-related genes *MT-ND5* and *MTRNR2L12* - reported to be involved in neuronal energy regulation[57–59] and neuronal survival respectively[60] - exhibited increased expression along maturation both in the 7 PCW tissue and in the day 70 and 120 MLOs.

To further map the distribution of specific developing patterns across spatial developmental trajectories, we identified two clusters within each sample and assigned identities based on maturation markers (*HES1*, *SOX2* and *VIM* in progenitors; *TH*, *DCX* and *MAP2* postmitotic neurons) (Fig. 5F). These clusters were confirmed at protein level with immunofluorescence analysis for SOX2 (progenitor) and MAP2 (post-mitotic neurons) (Supp Figs. 14 and 15). Their spatial distributions followed maturation paths in tissue and midbrain organoids. We investigated the expression of genes known to be involved in brain patterning during development[11] (Fig. 5G). We observed region-specific expression of *EN2*, *NKX2-2* and *WNT5A* at 7 PCW, with genes localized in the rostral ventral area of the developing midbrain. No specific gene expression in the VZ was observed at this timepoint. Instead, we detected the expression of *WNT5A* and *SHH* in the immature ventral compartment at 11 PCW and closer to aqueduct at 17 PCW samples. *EN2*, *STMN2*, *BARHL1* and *EMX2* showed diffuse expression in post-mitotic compartment across all timepoints analyzed. Similar distribution was observed for all genes across the immature and postmitotic compartments in midbrain organoids.

We then investigated the spatial development of mDA neurons. As we detected strong, spatially correlated expression of *CALB1* and *ALDH1A1* only at 17 PCW, we thus focused on this timepoint. We first manually annotated four regions according to the expression of *TH* in spatial transcriptomics, immunofluorescence images and the anatomy of the fetal brain (Fig. 6A, B). We computed differentially expressed genes (DEGs) between all pairs of regions and found that regions 1 and 2 exhibited similar expression patterns, as did regions 3 and 4. Regions 1 and 2 expressed high levels of dopaminergic-related genes (*EN1*, *TH*, *DDC*) and neuronal development markers (*TUBA1A*, *NNAT*)[61,62]. We also detected expression of serotonergic markers *SLC6A4*, *FEV*, as were recently observed in scRNA-seq of mDA neurons[18], suggesting early molecular heterogeneity within mDA populations. Regions 3 and 4 showed increased expression of a set of mitochondrial related genes (*MT-CO1*, *MT-ND1*, *MT-ATP6*, *MT-ND4L*, *MT-ATP8*), suggesting increased metabolic activity along maturation. Region 4 showed the most mature phenotype, with expression of the neuronal activity-

related genes *SLC17A6*[63], *NRN1*[64], and *ADCY1*[65] (Fig. 6C). To understand which genes were associated with the maturation processes of the SN and VTA, we tracked the expression of DE genes in those spots expressing *CALB1* and *ALDH1A1*, in addition to TH (Fig. 6D).

We then investigated ligand-receptor interactions in neighboring spots to highlight the activation of communication pathways during ventral midbrain development and evaluate their preservation in MLOs. We used COMMOT[66] to compute the expression of pathways at each spot and then used the Moran's I non-parametric test for spatial autocorrelation to identify those with a large spatial component to their expression. As a result, we identified highly active communication pathways at 11 PCW fetal midbrain (Fig. 7A; Supp Fig. 18). Among the 20 most highly activated ligand-receptor pathways in this tissue, we identified an important role for heparin binding growth factors, involved in neuronal maturation. In 11 PCW fetal midbrain, the ligands, *PTN*[67,68] and *MDK*[69,70], showed expression in the VZ and IZ, throughout the ventral medial axes and the dorsal tectum. Both growth factors have a common receptor, the protein tyrosine phosphatase receptor type Z1 (*PTPRZ1*)[71], which was expressed along the progenitor area and ventral medial line. We did not detect *PTPRZ1* in the dorsal tectum of the midbrain (Fig. 7A). A small number of neighboring ligand-receptors pairs were identified in MLOs (Fig. 7B). We then confirmed expression of PTPRZ1 in the ventral area of the midbrain, whereas it was absent in the dorsal area of the midbrain (Fig. 7C). This highlights PTPRZ1's role in the specification of ventral progenitor development and fate decision. Strong co-localization was also found for *MDK* and another of its receptors *LPR1*[2], mostly expressed in both the ventral and lateral IZ of the developing midbrain at 11 PCW. High expression of *MDK/LPR1* ligand-receptor pair was also observed in the MLOs across differentiation time points (Supp Fig. 18).

Finally, we used the resolution provided by spatial transcriptomics to quantify the spatial similarity between fetal tissue and organoid samples to further elucidate temporal dynamics. Organoid models could indeed allow better integration of signals from neighboring cells, in a manner similar to what occurs in vivo. However, the extent to which these local cellular interactions can be faithfully replicated in midbrain organoids remains to be determined[26]. To provide an evaluation of this, we first labelled each spot according to the cell type that was estimated in the deconvolution to be most abundant (Fig. 7D). Then, for each sample, we quantified the spatial relationships between these labelled cell types (Fig. 7E). We found the organoids to share high spatial similarities with fetal tissue that align along maturation. The d40 organoid displayed some similarity to the 7 and 11 PCW tissues, and reduced similarity to the 17 PCW. The d70 and d120 organoids showed very low similarity to 7 PCW, low similarity to 11 PCW, and medium similarity to 17 PCW. The d40 organoid showed low similarity to d70 and d120, whilst these later timepoints exhibited medium similarity to one another. As expected, tissues showed less similarity as the differences in their timepoints increased. We estimated the temporal alignment of organoids and tissue timepoints and found that the d40 organoid corresponded to a tissue timepoint of slightly over 11 PCW whilst the d70 and d120 organoids both corresponded to a tissue timepoint of approximately 15 PCW (Fig. 7F). Taken together, these data highlight the parallel spatial development of midbrain dopaminergic neuronal populations in fetal tissue and MLOs across time.

## A human-derived mDA neuronal in vitro model allows temporal and cell-type specific analysis of dynamics in neurological disease

Our previous observations provide further evidence of temporal variances in MLOs and fetal midbrain development. This poses the question of whether the relative maturity of MLOs may present potential limitations in their use for disease modeling. To address this paradigm, we investigated dopaminergic-specific temporal disease

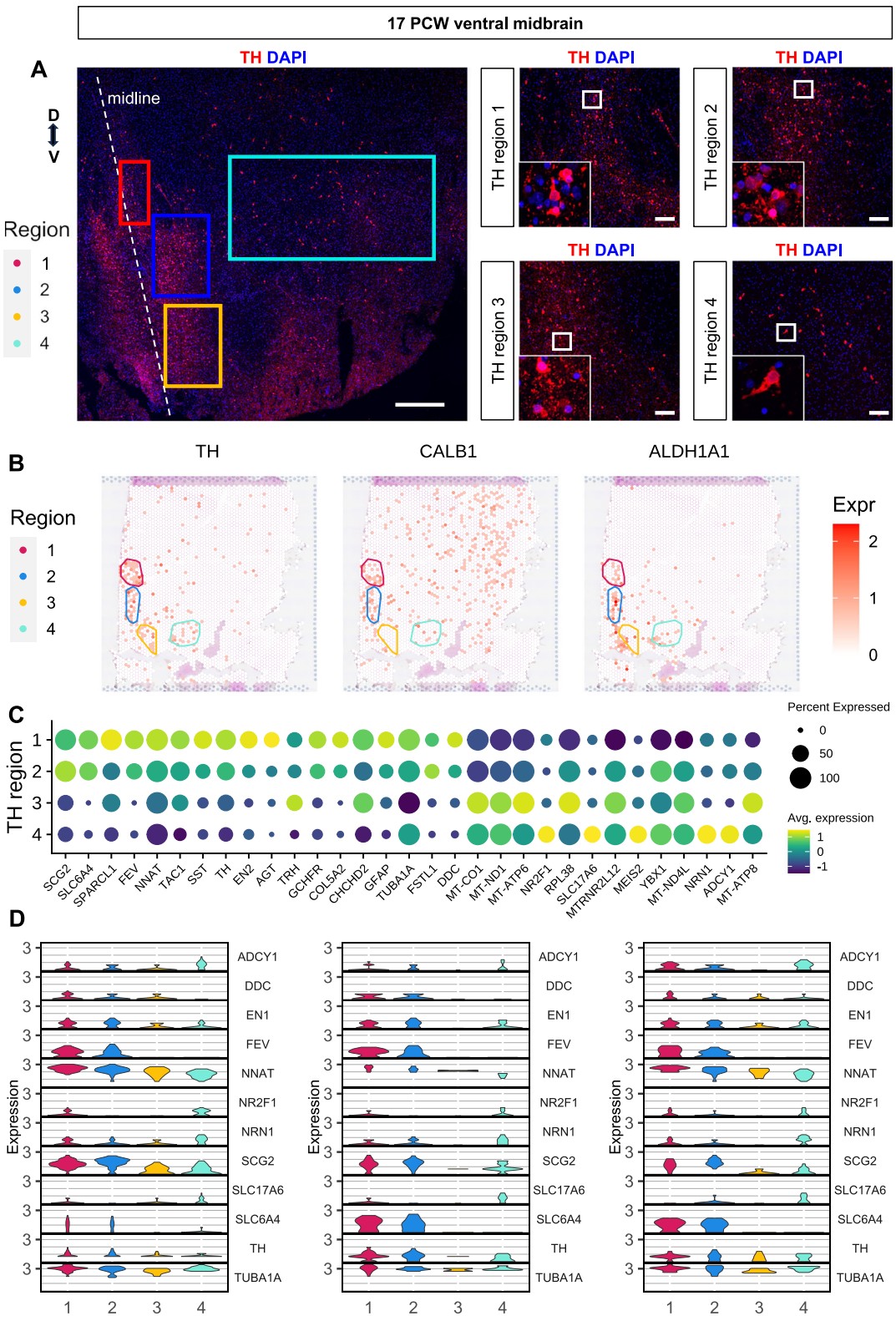

**Fig. 6 | Spatially defined maturation of dopaminergic neurons at 17 PCW.**
**A** Immunofluorescence analysis of 17 PCW human ventral midbrain for TH. Squares indicate different anatomical regions of TH expression across the ventral midbrain, including the midline and lateral areas. Nuclei are stained with DAPI. Scale bars = 400 µm (*left panel*); 50 µm (*right panel*). **B** Regions associated with *TH* expression in 17 PCW tissue, overlaid over expression of markers of ventral midbrain dopaminergic subtypes *CALB1* and *ALDH1A1*. **C** Expression of genes associated different TH regions. **D** Expression of genes associated with dopaminergic neurons maturation and specification shown across spots positive for TH, CALB1 or ALDH1A1, split by the TH regions shown in (**B**). Each y axis starts at 0.

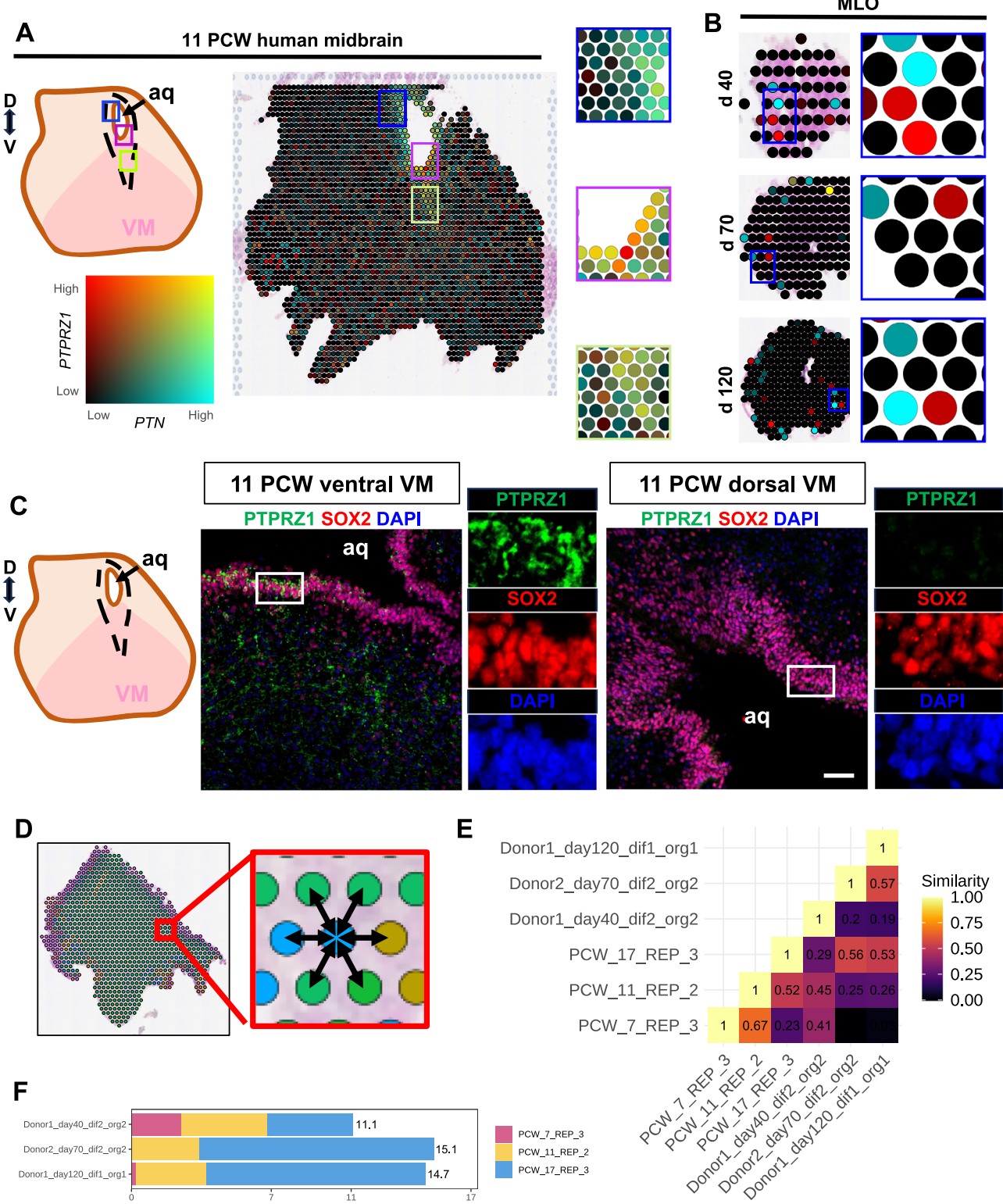

**Fig. 7 | Fetal and MLO midbrain systems share spatial cell-cell communication pathways and spatial characteristics. A** Expression in 11 PCW tissue of the PTN-PTPRZ1 ligand-receptor pair, identified by COMMOT to be highly active and shown by Moran's I to be spatially correlated. *Created in BioRender. BUDINGER, D. (2025)* https://BioRender.com/3akidui. **B** Expression in organoids of the PTN-PTPRZ1 ligand receptor pair. **C** Immunofluorescence analysis of 11 PCW human fetal coronal section for PTPRZ1 and SOX2. Nuclei are stained with DAPI. Scale bar = 100 μm.

Cerebral aqueduct (aq). *Created in BioRender. BUDINGER, D. (2025)* https://BioRender.com/gfxtdrp. **D** Schematic of similarity quantification approach: cell types of neighboring spots are counted and used to compute similarity between samples. **E** Quantification of spatial similarity between tissues and organoids. **F** Temporal alignment of organoids and tissues, where the timepoints of the tissues are averaged and weighted according to the organoid's spatial similarity to each one.

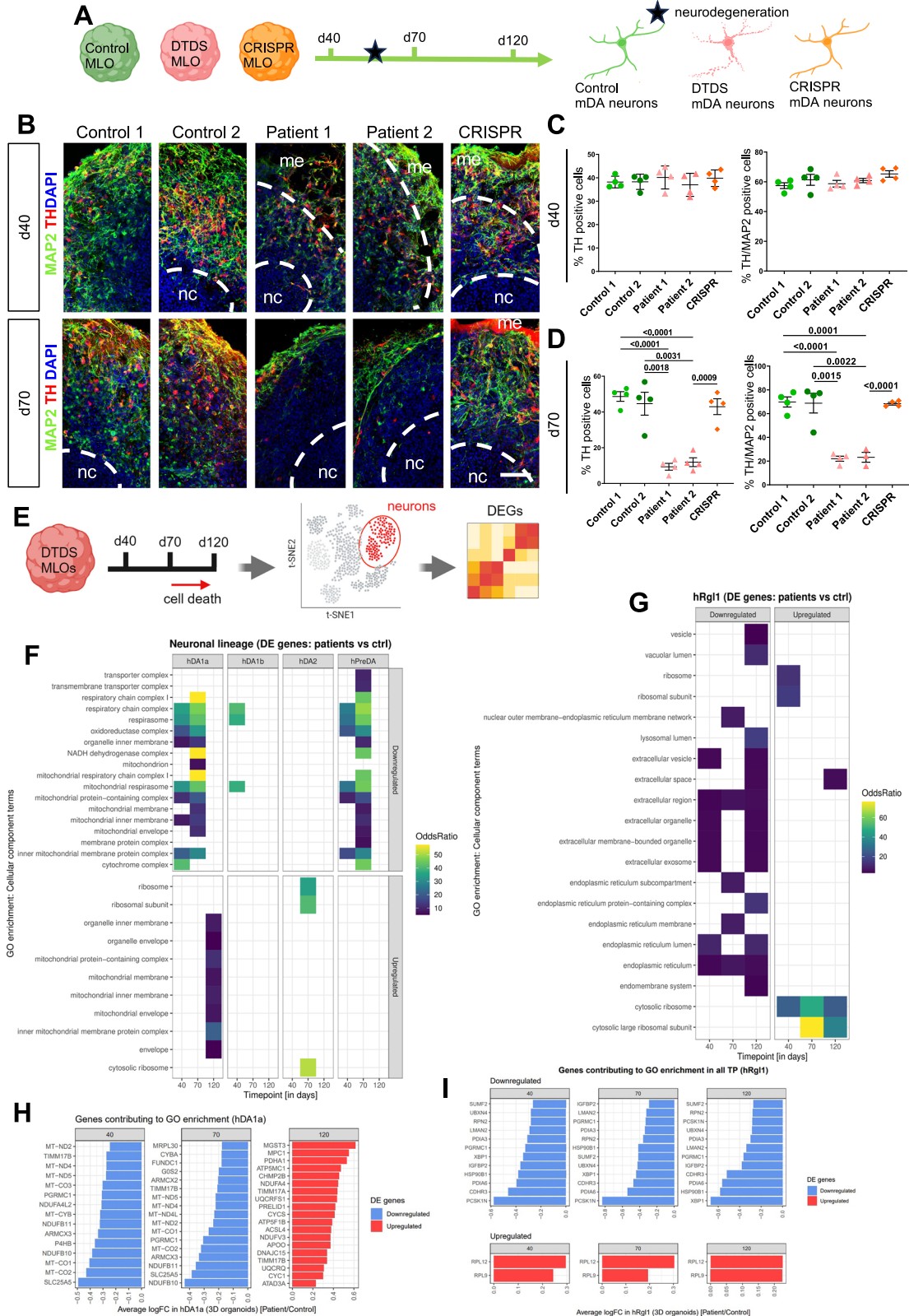

progression in an MLO model of the neurodegenerative disease, Dopamine Transporter Deficiency Syndrome (DTDS). DTDS is due to biallelic mutations in the *SLC6A3* gene, leading to impaired dopamine transporter[72] activity[73]. Using a 2D patient-derived mDA culture system, we have previously shown that DTDS is characterized by apoptotic neurodegeneration associated with TNFα-mediated inflammation, and dopamine toxicity[36]. We differentiated our

previously characterized DTDS patient iPSC lines together with two age-matched controls lines and isogenic CRISPR-corrected line[36] into MLOs (n = 4 donors) (Fig. 8A). After 20 days of differentiation, both the control and patient lines showed expression of midbrain precursors genes and proteins (Supp Fig. 19A, B). Derived MLOs were further matured over 40, 70 and 120 days in culture, and showed gene and protein expression profiles of mature mDA neurons (Supp Fig. 19A;

**Fig. 8 | MLOs derived from DTDS iPSCs allow investigation of disease progression in dopaminergic neurons. A** Experimental design for the analysis of DTDS-derived MLOs at different maturation time points (40, 70, 120 days of differentiation) to investigate progressive neurodegeneration. *Created in BioRender. BUDINGER, D. (2025)* https://BioRender.com/wvn03ts. **B** Immunofluorescence analysis for the neuronal markers TH and MAP2 at d40 and d70 of differentiation in Control 1, Control 2, Patient 1, Patient 2 and CRISPR isogenic-derived MLOs. Nuclei were counterstained with DAPI. Scale bar= 50 μm. **C, D** Quantification of TH positive and TH/MAP2 double-positive neurons for controls and patient lines (n = 4 for every line). Error bars indicate SEM. DTDS lines were independently compared to controls using a two-tailed Student's *t*-test for all analyses. Statistically significant differences indicate comparison of Control 1 or 2 to Patient 1 or 2 and Patient 2 to CRISPR. **E,** Experimental design for the differentially expressed genes (DEGs) analysis in mDA clusters. *Created in BioRender. BUDINGER, D. (2025)* https://BioRender.com/oo6rdkc. **F, G** Gene ontology enrichment analysis (GO) on cell compartments based on the DEGs in neuronal lineage and hRgl1 clusters, respectively, in patients compared to controls. **H, I** List of genes contributing to GO enrichment in hDA1 and hRgl1 clusters, respectively. Error bars indicate SEM. DTDS lines were independently compared to controls using a two-tailed Student's *t*-test for all analyses. Statistically significant differences indicate comparison of Control 1 or 2 to Patient 1 or 2 and Patient 2 to CRISPR. Necrotic core (nc), Matrigel embedding (me). Source data are provided as a Source Data file.

Supp Figs. 20–22). We detected a GABA/TH double positive neuronal population, as similarly observed in the human fetal midbrain tissue. We could also detect GFAP positive cells at later stage of maturation (120 days of differentiation).

Temporal analysis of DTDS patient-derived MLOs (day 40, 70, 120) revealed a significant decrease in total neuron count in patient-derived MLOs, compared to both control and isogenic control lines (Fig. 8B–D) after 70 days in culture. This neuronal loss was particularly evident in TH+ neurons, at 70 and 120 days of differentiation (Supp Fig. 23A–C), while no neuronal loss was observed in control and patient-derived MLOs at day 40. We also observed reduced total TH protein in DTDS MLOs at 70 days of differentiation despite expression of DAT, as previously reported in 2D mDA neuronal cultures (Supp Fig. 23D, E)[36]. Although single-cell transcriptomics did not reveal significant shifts in overall cell type composition between patient- and control-derived MLOs at any time point, we observed a trend toward progressive reduction of mature and immature neurons in patients. To elucidate the mechanisms associated with dopaminergic neuronal loss in patient MLOs, we analyzed apoptotic processes via cleaved Caspase3 (c-CASP3) immunofluorescence. We observed increased c-CASP3 positive cells in the patient MLOs and specifically in TH-positive neurons, confirming our previous study in 2D mDA patient-derived model (Supp Fig. 24A, B)[36]. Notably, this analysis excluded the necrotic core of MLOs.

Using our single cell dataset, we computed the differentially expressed genes (DEGs) between control and patient lines within each cell population identified in the MLOs, using Seurat (Fig. 8E, Supp Data S11). Our analysis focused into the dopaminergic neuron lineage and a subtype of radial glia cells (Rgl1). Gene enrichment analysis showed a downregulation of the cellular components related to the mitochondrial membrane and mitochondrial respirasome in hDA1a and hPreDA populations from patient MLOs at both 40 and 70 days of differentiation (Fig. 8F; Supp Fig. 24D). Those components were mainly associated with the mitochondrial electron transfer chain complex I and IV for both hPreDA and hDA1a (*MT-ND2, MT-ND3, MT-ND5, MT-CO1, MT-CO2, & MT-CO3*) (Fig. 8H; Supp Fig. 24C, D).

Interestingly, upregulated DEGs in hDA1a at day 120 are enriched in mitochondria components, and mainly associated to mitochondrial inner membrane (*MPC1, ATP5F1B, TIMM17B, TIMM17A, ATP5MC1*), cytochrome c (*UQCRFS1, CYC1, CYCS*) and cytoprotection (*PRELID1*), while ribosomal components are upregulated in hDA2 at day 70. (Fig. 8F, H). In the hRgl1 subtype, downregulated DEGs were mainly associated with the extracellular space and endoplasmic reticulum across all MLO differentiation stages, while upregulated genes were linked to the ribosome (Fig. 8G–I). Overall, these data show the temporal dynamics of DTDS disease progression, characterized by mitochondrial dysfunction specific to dopaminergic neurons prior to their neurodegeneration.

## Discussion

In this study, we provide a detailed molecular and cellular profiling of the first two trimesters of human fetal midbrain development from 6 to 22 PCW. Due to the rarity of these late-stage embryonic samples, our study offers the opportunity to provide both transcriptomics (single cell and spatial transcriptomics) and immunofluorescence images as a valuable resource for the research community. Our analysis complemented by existing human datasets of midbrain development[6,7,11], reveals the structural and cellular complexity of the human midbrain, which emerges mostly during the second gestational trimester.

Indeed, during the second trimester of human fetal midbrain development, we observed the spatiotemporal emergence of major ventral midbrain nuclei, including the red nucleus (RN), substantia nigra[2], and ventral tegmental area (VTA)[74], along with the appearance of markers indicative of cellular maturity. Cellular maturation of mDA neurons occurs throughout radial migration of neuronal precursors from the ventricular zone to the mantle zone and then a tangential migration towards the specific nuclei location[9,23]. Our observations suggest that while radial migration of mDA neurons has occurred by 12 PCW, significant tangential migration, neuronal morphological maturity, and neuronal network complexity develop after this stage, mostly between 19 and 22 PCW. This reflects a late specification of dopaminergic nuclei similar to what is observed in mice, where the tangential migration and maturation of mDA neurons occur between embryonic days E13.5 and E18[6,22,75,76]. In agreement, histological analysis of second trimester fetal midbrain already reveals structural complexity similar to that of adult tissue[34,35]. In contrast, our transcriptomic analysis showed a discrepancy between second trimester and adult samples. It is likely that major changes in midbrain maturation happen later in human development which, given the limitations in acquiring human fetal samples, have not been captured in our analysis[11].

We also investigated the similarity of later stage human midbrain development to human pluripotent stem cell-derived midbrain dopaminergic neuronal models, including 3D models[7,8,18]. To our knowledge, there is limited understanding as to whether human pluripotent stem cell-derived mDA models truly replicate cell type location, proportion, gene expression, cell-to-cell communication and differentiation along developmental time and space of the human midbrain. Therefore, understanding the extent of biological recapitulation in these models is essential to justify their use as biological proxies.

To evaluate the alignment of models at transcriptional level, we performed an integration of our dataset with scRNA-seq data from fetal, post-mortem adult, and iPSC-derived in vitro midbrain models (La Manno et al., 2016; Agarwal et al., 2020; Fiorenzano et al., 2021; Birtele et al., 2022; Zagare et al., 2022). This generated a unified cellular profiling across developmental stages and platforms. We observed a shared developmental axis, with in vitro models aligning with early first-trimester profiles and diverging from in vivo development around 10 PCW.

To further investigate this, we specifically examined the dopaminergic cell population. Comparisons of hDA2 clusters between fetal samples and MLOs showed higher expression of synaptic genes in vivo, suggesting that in vitro models lack appropriate synaptic targets or sufficient maturity in vitro. This discrepancy underscores the difficulty to fully recapitulate developmental progression in vitro, raising

questions regarding the utility of organoids in modeling later-stage diseases such as juvenile-onset, adult-onset or sporadic forms of parkinsonism. Yet, our immunohistochemistry analyses revealed that GIRK2, a potassium channel typically expressed in mature DA neurons (Reyes et al., 2012), localized to neurites in both late fetal samples (22 PCW) and day 120 MLOs. This suggests that certain neuronal features, such as axonal localization of functional proteins, may still reach advanced maturation in vitro despite the transcriptomic resemblance to earlier stages.

Using the single-cell profiles of the first and second trimester, we reconstructed astrocytic and neuronal trajectories rooted in early midbrain progenitors. Pseudotime analysis revealed gene expression dynamics across development, with patterning genes (e.g., *ID4*, *TTYH1* and *VIM*) being upregulated first, and neuronal maturation markers (e.g., *STMN1/2*, *SYT1*, *NSG2*) at the end. In the astrocyte lineage, key transporters and neuroprotective genes (e.g., *SLC1A2*, *SLC6A11*, *AQP4*, *CST3*) emerged late, indicating functional maturation. Interestingly, pseudotime progression in this lineage correlated strongly with gestational age, supporting the temporal resolution of our approach.

A known limitation of scRNA-seq for profiling the developing brain is the underrepresentation of fragile, highly arborized neurons, such as DA neurons, particularly at later developmental stages, a bias also observed in previous studies[11]. While snRNA-seq may better preserve these populations, we chose scRNA-seq because it retains cytoplasmic transcripts and mitochondrial gene expression, which are essential for studying neurodegenerative and mitochondrial-related diseases such as DTDS[36]. Additionally, this approach enabled direct comparison with other fetal midbrain scRNA-seq datasets[6,7,11], thereby minimizing cross-platform differences. Despite reduced neuronal capturing yield, we were still able to identify a downregulation of the mitochondrial respirasome in patient-derived MLO with DTDS.

We hypothesized that some discrepancies in maturity between fetal midbrain and MLOs may be attributed to the loss of spatial and cell-to-cell interactions inherent in single cell approaches, which decontextualize cell behavior. Spatial transcriptomics addresses this limitation by spatially aligning the RNA content of cells, enabling the comparison of in vitro models and tissue samples based on spatial organization and cellular interactions[77]. Our spatial analysis uncovers developmental pathways both in vitro and in vivo, enabling us to identify conserved DEGs along these maturation pathways and to find key genes associated with the maturation of different classes of dopaminergic nuclei. Importantly, we observed upregulation of genes related to synaptic neuronal activity and mitochondrial function along the dopaminergic developmental axis, likely reflecting the increasing metabolic demands associated with neuronal maturation, synaptic readiness, and neurotransmission[78,79]. The mitochondrial trajectory may support the establishment of DA neuron subtype identity while also increasing their susceptibility to mitochondrial dysfunction-related degeneration, offering a developmental explanation for their selective vulnerability in neurodegenerative diseases[80,81]. These molecular profiles may help elucidate the genetic programs driving functional maturation and subtype specification within the mDA system, and they provide a framework for further evaluating the fidelity of organoid models in reproducing human midbrain development.

Additionally, our spatial analyses revealed important cell-to-cell communication networks within the tissue. We identified active ligand-receptor pathways, notably involving heparin-binding growth factors *PTN* and *MDK* and their receptor *PTPRZ1*, conserved both in vivo and in vitro systems. We also used spatial transcriptomics to quantify spatiotemporal similarity between midbrain tissues and in vitro models. Notably, this revealed an increased similarity between late-stage MLOs and later stages of fetal midbrain compared to what we observed with scRNA-seq, aligning day 70 and 120 MLOs closer to fetal midbrain samples from the second trimester of gestation, both peaking at approximately 15 PCW. This underscores the importance of our spatial analysis, highlighting the importance of incorporating the spatial context when considering the in vitro recapitulation of developmental progress.

Although iPSC-derived models may not fully replicate the exact maturation stage of their in vivo counterparts, their usefulness may largely rely on the specific cell type and phenotype under study. Indeed, despite certain imperfections, human-derived in vitro models offer the possibility to investigate disease progression over time in a patient relevant context. We leveraged this by developing MLOs from DTDS patient-derived iPSCs and investigating neurodegenerative progression over time at molecular and single cell transcriptomic levels. Interestingly, we observed specific dopaminergic neurons loss, which was not detected in our previous study in 2D mDA cultures[36]. This might reflect the ability of MLOs to better recapitulate the pathological cellular interactions and microenvironment of the midbrain in a diseased state[82,83]. Temporal scRNA-seq transcriptomic analysis in DTDS MLOs indicates mitochondrial dopaminergic-specific dysfunction at an early stage of maturation, before dopaminergic neuron loss occurs. Notably, DTDS MLOs at a late stage of maturation showed dysregulation of cytochrome-related genes and increased expression of *PRELID1*, which might indicate a progression of mitochondrial dysfunction and neurodegeneration over time[84–86]. Overall, our analysis suggests that DTDS primarily affects the dopaminergic neurons, at least in the early stages of the disease, with a process primarily impacting mitochondria. Understanding these neurodegenerative mechanisms, particularly the role of different cellular subtypes in the progression of DTDS, will aid in the development of disease specific therapies, as well as broader spectrum agents that may more generically target neurodegeneration.

In conclusion, our analysis provides a detailed characterization of the main cell types and tissue organization across the first two trimesters of human midbrain development, including the first reported profile of second trimester midbrain. Moreover, this study importantly highlights key similarities and differences between the human fetus and human-derived midbrain models, revealing the limits of in vitro models at late stages, while emphasizing the importance of spatial context in accurately benchmarking in vitro systems against in vivo development. Ultimately, our description of later-stage midbrain cell populations may enable future refinement of in vitro culture systems, better reflecting human dopaminergic and glial populations across late developmental stages, leading to improved modeling.

## Methods

### Human embryonic tissue collection

The human embryonic and fetal material were provided by the Joint MRC/Wellcome Trust (grant MR/R006237/1) Human Developmental Biology Resource (www.hdbr.org). Human fetal tissues were collected from 6 – 22 PCW from legally terminated embryos and gestational age of each embryonic and fetal was determined by either crown rum length (mm), foot length and/or knee-heel length (mm). Samples from HDBR Newcastle were shipped overnight on ice in Hibernate media (ThermoFisher Scientific), while samples from HDBR London were handed in DMEM medium and processed on the same day.

### Generation of iPSC-derived 2D mDA neurons and 3D MLO

2D mDA neurons and 3D MLOs were derived from iPSCs of two healthy individuals, two patient lines with missense mutations in *SLC6A3* (patient 1: c.1103 T > A, p.L368Q; patient 2: c.1184 C > T, p.P395L) and a CRISPR-Cas9 corrected isogenic line from patient 2[36]. IPSCs were differentiated into 2D mDA neuronal cultures as previously described[36,87]. For MLO generation, iPSCs at day 0 were dissociated to a single cell suspension, resuspended in embryoid body medium (EBM) and plated at a density of 10,000 cells/well in a 96 low-attachment U-shape wells plate (Corning). EBM consisted of 1:1 DMEM/F12:Neurobasal medium, 1:100 N2, 1:50 B27 w/o vitamin A supplement (Invitrogen), 1%

GlutaMax, 1% minimum essential media-nonessential amino acid (MEMNEAA), 100 U/ml penicillin G and 100 µg/ml streptomycin (Gibco), and ROCK-inhibitor (Thiazovivin) for the first two days. EBM was supplemented with: 0.1% β-mercaptoethanol (Gibco), 1 µg/ml heparin (Sigma-Aldrich), 10 µM SB431542 (Cambridge Bioscience), 100 nM LDN193189 (Generon), 0.8 µM CHIR99021 (Tocris Bioscience) and 100 ng/ml hSHH-C24-II (R&D Systems), and on day two, 0.5 µM purmorphamine (Cambridge Bioscience) was added. At day 7, MLOs were embedded in 10 ul Matrigel and SB431542 withdrawn from medium. The following day, embedded MLOs were transferred to a low attachment plate and cultured in final differentiation medium, consisting of Neurobasal medium, 1:100 N2 supplement, 1:50 B27 w/o vitamin A supplement, 1% GlutaMax, 1% MEMNEAA, 1% Pen/Strep, 10.1% β-mercaptoethanol (Gibco), 0.2 mM ascorbic acid (AA) (Sigma), 20 ng/ml BDNF (Miltenyi Biotech), 0.5 mM dibutyryl c-AMP (Sigma-Aldrich) and 20 ng/ml GDNF (Miltenyi Biotech), 200 ng/ml laminin (Sigma-Aldrich). From day 9 to day 14, medium was supplemented with 100 ng/mL FGF8 (Miltenyi Biotech), and with γ-secretase inhibitor DAPT (10 µM, Tocris) from day 30 until the end of differentiation. The organoids were cultured under static conditions and medium changed every other day.

## Immunostaining

Fetal midbrain samples at 7, 12, and 16 PCW used for immuno-fluorescence analysis were obtained from the HDBR after paraffin embedding. Samples were sectioned at 8 µm using a microtome. Antigen retrieval was performed prior to immunostaining in pH 6 citrate buffer. Other fetal midbrain samples were washed in ice-cold PBS immediately upon delivery, and meninges were carefully removed using tweezers. Samples were fixed in ice-cold 4% paraformaldehyde (PFA) for 10 min (2D mDA neurons) and in 4% PFA overnight at 4 °C (MLOs, fetal samples 6 and 7 PCW) or 72 hour at 4 °C (fetal samples 11, 12, 16, 17, 19, 22 PCW), washed with PBS, and fetal and MLO samples immersed in 30% sucrose in PBS at 4 °C until saturated. Fetal midbrain and MLO samples were embedded in cryomolds using optimal temperature cutting (OCT) compound and frozen at −80 °C. 2D mDA neurons, cultured on LabTek slides, were stored in 1 X PBS at 4 °C. All embedded samples were sectioned at 12 µm onto SuperFrost Plus microscope slides. All slides were washed with 1 X PBS and blocked using 5% bovine serum albumin (BSA), 0.3% Triton in 1 X PBS for 1 h at room temperature. Primary antibodies (Supp Data S12) were diluted in blocking solution and incubated overnight at 4 °C, followed by 3 washes in 1 X PBS. Secondary antibodies were diluted in blocking solution and incubated 1 h at RT, followed by 3 washes in 1 X PBS. All secondary antibodies used were AlexaFluor used at a dilution of 1:400.

Imaging was performed using a Zeiss LSM710 confocal microscope. Image stacks were acquired using either a 10x or 20x objective, with a z-step size of approximately 2.5 µm and a resolution of 1024 or 2048 pixels. Whole fetal samples were imaged using the mosaic function in the Zeiss confocal software and stitched using Zeiss software tools. Full size images at high resolution are available at https://hdbratlas.org/gene-expression/midbrain.html.

For quantification, 4 random fields were imaged from each independent experiment, and 1200 randomly selected nuclei were quantified using ImageJ software (National Institutes of Health).

## Reverse transcription and qPCR

Total RNA was extracted following RNeasy kit guidance (QIAGEN). RNA purification and cDNA preparation were performed according to manufacturers' guidance (Invitrogen). Samples cDNA were mixed in a qRT-PCR plate with MESA BLUE qPCR 2X MasterMix Plus for SYBR Assay (Eurogentec) that contained the appropriate target primer mix (ratio: 1 µl primers/10 µl MasterMix) (FOXA2, LMX1A, LMX1B, EN1, EN2, OTX2, NURR1, TH, PITX3, DAT, SNCA and GAPDH, Supp Data S13). Each target was plotted in triplicates for each biological sample. GAPDH was

used as a housekeeping gene for normalization purposes. qRT-PCR was performed on the StepOnePlusTM Real-Time PCR System (Applied Biosystems) with the following protocol: initiation at 95 °C for 5 min, 40 cycles (denaturation 95 °C for 15 s, annealing/elongation at 60 °C for 1 min$ elongation 72 °C. Gene expression was analyzed using the ΔΔCT method, with an age-matched control mDA line as control.

## Single cell isolation and 10X sequencing

Fetal, 3D and 2D iPSC-derived samples were mechanically dissociated to generate single cell suspensions using the Papain Dissociation System (Worthington). Following the supplier's recommendations, the EBSS was equilibrated with $O_2:CO_2$ to improve cell survival. Samples were weighted and minced into small pieces using open-wide tips coated in BSA 7.5%. After three ice-cold PBS washes, 200 uL of the dissociation mix (18.85 U/mL Papain, 50ug/mL DNase I, 1:4 Accutase/Accumax, 4ug/ml Actinomycin D) for 30 min at 37 °C on a thermo-shaker (750 rpm). Samples were triturated 10 X with a narrow pipette tip every 10 min for a total of 40 minutes. Cells were then pelleted at 300 g for 4 minutes and resuspended in the inhibitor mix (148.25uL EBSS, 50ug/mL DNase I, 17.5uL Ovomucoid for 175uL). 90 uL of Ovocumoid and 90 uL of EBSS were added on the top and gently mixed by tube flickering. Cells were pelleted at 300 g for 4 minutes. After discarding the supernatant, the pellet was resuspended in 10X buffer and filtered (40um cell strainer, membrane humidified with 10X buffer). Alive cell count target was >90% (LUNA cell counter, acridine orange/propidium iodide stain). Barcode and library preparation was performed using the 10X Genomics Chromium platform (v.2) and following the manufacturer's instructions. Briefly single cell suspensions were prepared and processed using the Chromium Single Cell 3' Reagent Kits v3 (10x Genomics, CG000183 Rev A). Cells were encapsulated into Gel Bead-In-EMulsions (GEMs), followed by reverse transcription, cDNA amplification, and library construction. Samples with different genetic backgrounds were randomly pooled across 10x libraries (ie GEM wells), as indicated in Supp Data S14. Nine libraries (2D and 3D iPSC-derived samples) were sequenced with Illumina-HTP HiSeq 4000 paired-end sequencing. Four libraries corresponding to the fetal samples were sequenced in a separate batch using Illumina-HTP NovaSeq 6000 paired-end, across 4 lanes. To discard any potential sequencing lane effect, each 10x library was sequenced in all 4 lanes. Specifically, each fetal library was further subdivided into four samples, with each sample sequenced in one of the 4 lanes of the NovaSeq600 paired-end sequencing.

## Data preprocessing (single-cell RNA-seq)

We processed the thirteen pooled 10x libraries (ie. GEM well) using CellRanger (version 3.1.0) (Supp Data S2, S14; Supp Fig. 10A). Four 10x libraries were generated for fetal samples, while nine libraries were generated for the in vitro models (2D and 3D), three per differentiation timepoint (D40, D70 and D120). For fetal samples, donors from either the first trimester (PCW10, PCW12) or second trimester (PCW16, PCW20) were pooled together, and two technical replicates were generated per pool. For in vitro samples, we randomized the pool composition to include differentiations from four donors at a time, either modeled in 2D or 3D. We avoided mixing samples with the same genetic background, either generated from different models (ie. Control1-2D with Control1-3D), or corrected from a patient-derived line (ie. Crispr1-3D with Patient2-3D). This strategy enables unambiguous deconvolution of donor identity later on. Overall, each donor/model combination is represented in at least two 10x libraries (range: 2-9 replicates).

Using the "cellranger count" pipeline and the sequencing data (FASTQ files) as input, we performed alignment to GRCh38 reference genome, barcode counting and UMI counting to generate filtered feature-barcode matrices for each library. The genes counts were quantified using the Ensembl 93 reference gene annotation (N = 33,538

genes). For each 10x library, we used 32 CPU cores, 200 GB of memory and a maximum of 64 jobs to be run simultaneously. Only for fetal samples, the "-fastqs" argument was used to specify the paths to the multiple FASTQ files generated from the library subdivisions in the NovaSeq600 sequencing.

The matrices of each 10x library went through a quality control step based on the distribution of i) expressed genes per cell, and ii) the mitochondrial count fraction, so as to remove dying cells or those with broken membranes. For 2D and 3D-iPSC derived samples, cells expressing <1500 or >6000 genes were discarded, as well those with an excess of mitochondrial count fraction (>10%). For fetal samples, with a different distribution for both parameters, cells were removed when expressing <1500 or >5000 genes, and when the mitochondrial count exceeded 25%.

To remove doublets/multiplets in the single-cell libraries, we performed cell deconvolution with demuxlet[41], using existing genetic variation from the eight donors previously genotyped with SNP arrays (VCF file, File S1,S2). Demuxlet was run using a default prior doublet rate of 0.5 and only those cells unambiguously linked to a donor (singletons) were retained. All pooled libraries presented an acceptable singleton rate (>60%), and all expected donors were deconvoluted. After quality control and processing, each donor/model combination had between 1,346 and 9,394 cells profiled.

### Normalization, dimensional reduction, clustering and projection (single-cell RNA-seq)

After preprocessing, all 10x libraries were merged and all genes that were not expressed in at least 0.1% of the total cells were removed. The gene counts were initially normalized to a scale factor of 10,000 and log-transformed (log1p). We then selected the 2000 highly variable genes (HVG) based on the variance stabilizing transformation and scaled the data using default Seurat parameters (scale.max=10). Based on this selection, we then calculated the first 50 principal components (PCs) and batch-corrected them with Harmony (v.0.1.0)[43], treating each 10x library as a different batch (default parameters, tau=0, max.iter.harmony=10, max.iter.cluster=20). We used the batch-transformed PCs to compute a neighborhood graph (neighbors=10), visualize the graph using UMAP and performed cell clustering using the Louvain algorithm (resolution=0.75), identifying 24 different clusters (File S3). We used the Seurat R package (v.4.1.3)[42] for all the processing steps.

### Cell type annotation (single-cell RNA-seq)

To capture both general and specific cell identities, we created a two-tiered cell type annotation system: one based on the high-resolution clustering, and the other by grouping those subtypes into broader categories for low-resolution annotation.

In the high-resolution clustering, the annotation was performed using a manually curated set of markers (Supp Data S3) established in previous studies[6,18]. Initially, we iteratively performed differential expression (DE) for each unannotated Seurat cluster against all other clusters to generate a list of specific cell type markers. We identified cell types by intersecting the DE list with the curated markers, while also considering the expression level of each marker. We confidently identified 23 out of the 24 clusters, with one unknown cluster likely having a connective tissue identity. Based on those markers, we also calculated gene module scores per cell type to evaluate the specificity of the annotation using the "AddModuleScore()" function from the Seurat package (ie. Supp Fig. 9B).

In the low-resolution annotation, we grouped cells into 10 broad categories that reflect major developmental trajectories (ie. neurons, astrocytes, microglia, perivascular cells) and progenitor stages (early, late, neuronal), using a low-resolution clustering (resolution=0.1) as a guide (Supp Data S3).

### Differential abundance analysis (single-cell RNA-seq)

To assess cell type composition differences between models, we used the *miloR* package (v.1.2.0)[46] that performs differential abundance testing. This statistical framework assigns cells to neighborhoods based on a k-Nearest Neighbor (kNN) graph, being particularly well-suited for continuous trajectories observed in neurodevelopment or in differentiation.

We tested four different comparisons:
a) In vitro models: 2D vs 3D (accounting for differentiation day as a covariate)
b) In vitro 3D model vs in vivo fetal tissue
c) Post-conceptional weeks of fetal gestation (early vs late fetal)
d) In vitro 3D models of DTDS: patient- versus derived-control lines

For each comparison, the kNN graph was built using the *buildGraph()* function, using 30 nearest-neighbors (k = 30) and 30 Harmony batch-corrected dimensions (d = 30). Then, we defined neighborhoods on a graph with *makeNhoods()* function using the same parameters (k = 30, d = 30) and randomly sampling 30% of the graph vertices. We only used 20% of the vertices in the fetal comparison. Cells were then counted for each neighborhood and experimental sample: for "a-b" comparisons, the experimental sample is defined by combining "donorDimensions" and "timePoint" metadata columns from File S3; for the "c" comparison, is defined by combining "midBrainId" and "timePoint" columns, and for the "d" comparison is defined by combining the "donorId" and "timePoint" columns. After that, distances between neighborhoods were calculated using the *calcNhoodDistance()* function using again 30 Harmony dimensions (d = 30). We performed differential neighborhood abundance testing with the following design for each comparison:
a) ~ Time + originDimensions (discrete: 2D vs 3D)
b) ~ originDimensions (discrete: 3D vs fetal)
c) ~ timePoint (continuous: post-conceptional weeks 10, 12, 16, 20)
d) Two designs were tested, either taking the differentiation day as a covariate (~ Day + Condition), or within each differentiation timepoint (~ Condition) for both the high- and low-resolution annotation.

Finally, neighborhoods were annotated with a given cell type when the most frequent identity among the cells from each neighborhood was >70% (>75% in "c" comparison). Otherwise, non-homogenous neighborhoods were annotated as "Mixed". Results were visualized with a custom beeswarm plot using default FDR significance level (<0.1). The log-fold change values of differentially abundant cell type-specific neighborhoods indicates the following enrichments between conditions:
a) Enriched in 3D (logFC>0), enriched in 2D (logFC<0)
b) Enriched in fetal (logFC>0), enriched in 3D (logFC<0)
c) Enriched in late fetal (logFC>0), enriched in early fetal (logFC<0)
d) Enriched in DTDS patient-derived lines (logFC>0), enriched in controls (logFC<0).

### Differential expression analysis (single-cell RNA-seq)

We performed differential gene expression (DGE) analysis between different conditions and experimental contexts:
1) Comparison between 3D and 2D in vitro models within differentially abundant dopaminergic populations (hDA1a, hDA1b, hDA2). To do so, we first aggregated neighborhoods in groups based on their cell type identity. We then ran differential expression using the *testDiffExp()* function of the miloR package for each dopaminergic population using the input design (~ Time + originDimensions). Differentially expressed genes were selected based on significance level (adjusted P-value < 0.05) and fold-change (upregulated: FC > 1.75; downregulated: FC < 1/1.75).

2) Comparison between fetal hDA2 and 3D hDA2 dopaminergic population. We subsetted our integrated dataset by only considering hDA2 cells from this publication and then filtering out 2D samples. Then, we set the object's identity class to the metadata column "system", and we ran differential expression by using the *FindMarkers*() function from Seurat, being the two comparing classes "Foetal" and "Organoid". Significance level was set at an adjusted P-value < 0.05 and a fold-change difference on expression of 1.5.

3) Patient versus controls (2×2 donor comparison in 3D models): For each timepoint (days 40, 70 and 120) and cell type with a minimum of 10 cells for both conditions, we ran differential expression analysis using the *FindMarkers*() function from Seurat. Significance level was set at an adjusted P-value < 0.05 and a fold-change difference on expression of either 1.5 (upregulation) or 1/1.5 (downregulation).

### Gene ontology enrichment analysis (single-cell RNA-seq)

Based on the differential expression analysis, we performed gene ontology enrichment analysis on the following results:

1) Comparison between 3D and 2D in vitro models within differentially abundant dopaminergic populations (hDA1a, hDA1b, hDA2). The gene universe was defined by the 22,032 genes from our single-cell data that passed QC on preprocessing. Then for each dopaminergic population, we run the functional enrichment on the list of DE genes using the *gost*() function from the *gprofiler2* R package[53]. We set the significance threshold at an adjusted P-value < 0.05 using the g:SCS method for multiple-test correction. We defined the gene universe in the "custom_bg" and used the functional annotation from the Cellular Component category of Gene Ontology (GO:CC)[88]. Based on the term size, the difference between the effective domain size and the term size, the query size and the intersection size, we computed the odds ratio for each term. In Fig. 3G, only the shared GO:CC enriched terms in dopaminergic populations hDA1a, hDA1b and hDA2 are shown.

2) Comparison between fetal hDA2 and 3D hDA2 dopaminergic population. The gene universe was defined by the 22,032 genes from our single-cell data that passed QC during preprocessing. Before the GO analysis, we annotated each gene with their corresponding Entrez gene identifiers using the package *org.Hs.eg.db* from R/Bioconductor. We filtered out those genes without correspondence or showing duplicate identifiers. We then run the hypergeometric test for GO term overrepresentation of biological processes (GO:BP) conditional to the hierarchical GO structure (package *GOstats* from R)[89]. The test was run separately for upregulated and downregulated genes. In each case, we used the following thresholds for significance: minimum gene set size = 25, maximum gene set size=500, adjusted P-value < 0.05, minimum number of counts per gene set = 10. In Fig. 3H and Supp Fig. 10C, only the top-5 upregulated biological processes and top-10 are shown, respectively. In Supp Fig. 10D, only the top-10 downregulated biological processes are shown.

3) Patient versus control (3D models): We performed GO enrichment analysis on the list of differentially expressed genes for the three dopaminergic populations (hDA1a, hDA1b, hDA2), dopaminergic precursors (hPreDA), dopaminergic neuroblasts (hNbDA), as well midbrain precursors (hMidPre) and radial glia type 1 (hRgl1), separately for upregulated and downregulated genes. We used the same parameters to run the *gost*() function and to compute the odds ratio as detailed in the first comparison. Results are illustrated in Fig. 8F,G. We then evaluated which genes contributed the most to the hDA1a enrichment on cellular components, either in downregulation (days 40/70) or in upregulation (days 120) (Fig. 8H). Additionally, we checked which genes consistently contributed to cellular components that were enriched across all timepoints in hRgl1 (Fig. 8I), either upregulated (cytosolic ribosome) or downregulated (endoplasmic reticulum, extracellular region). Finally, we also checked which genes contributed to the downregulation of the mitochondrial components at day 120 in midbrain precursors (hMidPre) (Supp Fig. 23D).

### Trajectory analysis

We performed trajectory analysis of neurons and astrocytes across the fetal cells in our dataset, using a modified approach based on a cortical brain organoid differentiation study[47].

i). Inference of lineage trajectories

We converted the Seurat RDS object (File S3) into an "h5ad" object to integrate it on the Scanpy workflow (version 1.10.4)[90]. We then clustered the cells using the Leiden algorithm at high-resolution (resolution=1) and constructed a PAGA graph, pruning out edges with low weights (w < 0.05). By fixing the start of the trajectory at the largest node of early midbrain progenitors (cluster 3, at resolution=1), we then computed the shortest path to either dopaminergic neurons (cluster 20 as endpoint) or to astrocytes (cluster 12 as endpoint). In the case of astrocytes, we forced the trajectory to go through late midbrain progenitors (cluster 0). In neurons, the inferred trajectory followed this cell type progression (early midbrain progenitors, late midbrain progenitors, neuron progenitors, immature neurons, mature neurons) (Supp Fig. 11B–E, J, K).

ii). Analysis of the developmental trajectories

For each inferred trajectory (neurons or astrocytes), we calculated the 2000 highly variable genes within each sample replicate (two replicates per timepoint). At each time point (PCW 10, 12, 16 and 20), we then calculated the intersection of highly variable genes (HVG) across the two replicates. The union of these intersected gene sets across all timepoints was used as the final HVG set for each trajectory (n = 1486 in astrocytes, n = 1,709 in neurons).

We next performed principal component analysis (PCA) on each trajectory using the respective HVG set to evaluate whether PC1 captured progression along differentiation. Specifically, we calculated the adjusted r-square between PC1 variance and annotated cell type labels (Supp Fig. 11G, H, M). Upon confirming that PC1 reflected the expected emergence of cell identities, we used PC1 loadings as a proxy for pseudotime and examined gene loadings at either end of the axis to identify genes associated with early or late stages of each trajectory (Fig. 3J, Supp Fig. 11I, O).

To complement this analysis, we computed the diffusion pseudotime (dpt) for each trajectory, anchoring the root at cells from cluster 3 (resolution=1) with the highest expression of *TUBB4B*. This cluster is the same used for defining the starting point of the astrocytic and neuronal trajectories. To explore how single cells distributed along pseudotime, we computed the kernel density for each timepoint and plotted the mean and standard deviation across replicates.

Finally, we modelled gene expression dynamics along each defined trajectory using *tradeSeq* R package[48] to identify lineage-specific developmental regulators. We run the fitGAM() function with 8 knots. For each lineage, we used the startVsEndTest () function to identify the HVG with significant expression changes between the start and end of the trajectory (adjusted p < 0.001, abs(logFC)>2). To improve the stability of the comparison at sparse trajectory extremes, we trimmed cells in the 1st and 99th diffusion pseudotime percentiles. Genes with significant start-end changes are listed in Supp Data S15 (neurons) and Supp Data S16 (astrocytes). We further characterized gene expression profiles across five evenly spaced pseudotime bins for a curated set of markers, including morphogens, transcription factors, enzymes, transporters and structural genes, among others (Fig. 3K, Supp Fig. 11P). For astrocytes, we focused on 10 representative genes: AQP4, GFAP, IL33, S100B, ALDH1L1, GJA1, HOPX, ZBTB20, SLC1A2 and

SLC6A11. For neurons, we examined 13 genes: LHX1, OTX2, DDC, NR4A2, TH, SLC18A2, GIRK2 (KCNJ6), EN1, EN2, ALDH1A1, FOXA2, SHH and WNT5A.

## Integration analysis

We integrated our single-cell dataset with six published midbrain development datasets (total of 194,521 cells) that either contained cells from in vivo fetal development or from iPSC-derived in vitro models (File S4). Here we shortly describe each dataset and where the data was obtained from:

**This work** (2D, 3D, Fetal): It encompasses 49,284 cells, containing four developmental stages of fetal development (one donor sampled at post-conceptional weeks 10, 12, 16 and 20, respectively) and iPSC-derived 2D and 3D models. 2D models have been profiled at two differentiation timepoints (days 40 and 70; n = 2 controls), and 3D models at three timepoints (days 40, 70, 120; n = 2 controls, n = 2 patients, n = 1 patient-corrected line). After annotation, 24 cell types were identified, with 19% of the cells being dopaminergic neurons. To download our data, check the specific data availability section.

**La Manno et al. 2017** (Fetal)[6]: It encompasses 1,977 cells sampled at six developmental stages of fetal development (gestational weeks 6, 7, 8, 9, 10 and 11). 26 cell types were identified in annotation, with 3 dopaminergic populations accounting for 6.2% of total cells. Raw counts were downloaded from the Gene Expression Omnibus (GEO) using the accession identifier GSE76381 at https://www.ncbi.nlm.nih.gov/geo/query/acc.cgi?acc=GSE76381. From the list of available files, we only processed the embryo molecule counts file.

**Braun et al. 2023** (Fetal)[11]: It encompasses 16,800 midbrain cells from 2 donors sampled at post-conceptional weeks 8 and 14, respectively. The selected cells are a subset from a larger atlas of the first trimester developing human brain. In the midbrain, 11 cell classes were identified, of which 41.4% of the cells were neurons. Raw counts were downloaded from the complete processed dataset "*HumanFetalBrainPool.h5*", previously deposited in the *Github* repository: https://github.com/linnarsson-lab/developing-human-brain. We subset the original matrix of counts by selecting only "Ventral midbrain" cells from the metadata "Tissue" column, and by selecting only the valid genes, as defined by the authors.

**Birtele et al. 2022** (Fetal)[7]: It encompasses 23,483 cells from four fetal ventral midbrain samples, sampled at three developmental timepoints: post-conceptional weeks 6 (MP06-hVM-6wks), 8 (MP03-hVM-8wks) and 11 (MP02-hVM-10-5wks and MP01-hVM-11-5wks, technical replicates). In those samples, 10 cell types were identified, of which 16.3% were dopaminergic neurons. Raw counts were downloaded from the Gene Expression Omnibus (GEO) using the accession identifier GSE192405 at https://www.ncbi.nlm.nih.gov/geo/query/acc.cgi?acc=GSE192405.

**Fiorenzano et al. 2021** (3D)[18]: It encompasses 91,034 organoid cells sampled at four differentiation timepoints: days 15 (n = 2 organoids), 30 (n = 5), 60 (n = 5), 90 (n = 2) and 120 (n = 6). In those 20 samples, 8 cell types were identified, of which 16% of the cells were dopaminergic neurons. Raw counts were downloaded from the Gene Expression Omnibus (GEO) using the accession identifier GSE168323 at https://www.ncbi.nlm.nih.gov/geo/query/acc.cgi?acc=GSE168323. Only the 20 standard organoid samples ("*standardorgday*") were merged for further analysis. Metadata containing cell type annotation was shared by the authors after request.

**Zagare et al. 2022** (3D)[8]: It contains 6000 organoid cells mimicking the human embryo ventral midbrain, sampled at two differentiation timepoints (days 35 and 70). Raw counts were downloaded from the Gene Expression Omnibus (GEO) using the accession identifier GSE133894 at https://www.ncbi.nlm.nih.gov/geo/query/acc.cgi?acc=GSE133894. The cell type annotation presented in the original publication was not available at the time of this integration. Only 2 wild-type samples were included for downstream analysis.

**Agarwal et al. 2020** (PostMortem)[49]: It contains 5943 cells from 7 post-mortem substantia nigra samples, of which two are technical replicates (n = 5 donors). 10 cell types were identified, of which only 1.2% were dopaminergic neurons. Raw counts were downloaded from the Gene Expression Omnibus (GEO) using the accession identifier GSE140231 at https://www.ncbi.nlm.nih.gov/geo/query/acc.cgi?acc=GSE140231. We merged the 7 samples profiling substania nigra (N). Metadata was downloaded from Supplementary Data 2 of the original publication.

For the integration, we used a later version of the Seurat package (v.5.0.0). We initially merged the raw counts of the seven datasets and normalized the data. Then, we found the highly variable features for each layer separately, using default parameters. After that, we scaled the data and calculated the first 50 principal components (PCs) and ran a fast implementation of the mutual nearest neighbors, FastMNN integration (package *SeuratWrappers, v.0.3.19*), using the *IntegrateLayers*() function and the previous dimensional reduction. We computed a neighbourhood graph, using 30 of the MNN embeddings, and performed cell clustering using the Louvain algorithm (resolution=0.75). We visualized the dimensional reduction with a UMAP projection using the same 30 MNN embeddings from the integration. After this, we harmonized the columns from the different datasets to have a coherent metadata. Linked to cell type annotation, we created two columns:

"**annotation_mixed**": This column mixed all the original annotations per publication in one column. Below you can find the columns names of the original metadata:

Our reference: "seurat_clusters_24_Annot"
La Manno et al.: "CellType"
Braun et al: "CellClass"
Birtele et al: "AnnotType"
Fiorenzano et al: "NamedClusters"
Agarwal et al: "Level_2_cell_type"

"**annotation_unified**": To compare cell type composition differences, we established a unified cell type annotation based on the published annotations of the datasets. You can find each cell type and the corresponding unified definition in Supp Data S6.

## Pseudo-bulk transformation and correlation analysis

Leveraging the final integrated object, we excluded all those cells that did not correspond to a wild-type control (i.e. patient iPSC lines or edited lines). After that, we selected the 26 samples for pseudobulk analysis, generated by combining each dataset with its corresponding developmental timepoints (for fetal) or differentiation days (for 2D and 3D). For each sample, we computed the average expression per gene, considering only the 2000 highly variable genes, and using the batch-corrected values from the FastMNN integration. We then computed the Pearson correlation of the pseudobulk expression between all pairwise combinations of in vitro (2D, 3D) and in vivo (fetal or postmortem) samples. The results from this analysis can be visualized in Fig. 4F heatmap, that presents an additional hierarchical clustering of rows and columns to highlight the transcriptional similarity between samples. For that visualization, we used the ComplexHeatmap R package (v2.18.0).

## Pseudotime analysis

We performed pseudotime inference using three different R packages. Initially, we filtered-out cells belonging to rare cell types (n < 50) based on the unified annotation of the integrated dataset. Additionally, cells from patient lines or edited lines were removed.

**Slingshot (v.2.10.0)**[50]: Trajectory inference was performed using the *slingshot*() function, leveraging the MNN dimensional reduction from the integration, with the starting point set at floor-plate progenitors (FPP). We then extracted the pseudotime values corresponding to the maximum weight for each cell and lineage, ranking these pseudotime values into cell positions across the trajectory to ensure comparability with other pseudotime inferences.

- **Destiny (v.3.16.0)**[51]: The integrated Seurat object was converted to a SingleCellExperiment object for compatibility. We then created a diffusion map of the cells using the MNN dimensional reduction to model differentiation-like dynamics with the *DiffusionMap*() function. The map includes the diffusion components (eigenvectors) and their importance (eigenvalues of the diffusion distance matrix). The first diffusion component was used as a proxy for pseudotime and cells were ranked for comparability. Additionally, we computed the diffusion pseudotime with the *DPT*() function, but found the temporal ordering was not better resolved and therefore not used in downstream analysis.

- **Monocle3 (v.1.2.9)**[91]: The integrated Seurat object was converted to a Monocle3 object using the *new_cell_data_set*() function to transfer the expression matrices, metadata and feature attributes. Additionally, the UMAP embeddings generated with the MNN dimensional reduction during the integration were also transferred. Cells were clustered with a resolution of $10^{-4}$ using the UMAP dimensionality reduction to define the partitions. We then learned the principal graph trajectory that cells follow through the UMAP space within each partition. Afterward, cells were ordered according to pseudotime on the learned trajectory, with the start node set within the population of floor plate progenitors (FPP). This pseudotime was later ranked for comparability with the other tools. Following pseudotime inference, we tested genes for differential expression across the pseudotime and selected those that significantly changed (q-val<0.05, Moran's I test). We then used the list of pseudotime-dependent genes to cluster them into modules that are co-expressed across cells. Using a random seed (123) and a Louvain resolution list ($10^{-6}$ to 0.1), we obtained 50 gene modules of varying gene size.

Then, we compared the pseudotime distribution of neurons (as defined by the unified annotation) among the developmental timepoints and the differentiation days of each dataset (Wilcoxon test). Further, we also compared the pseudotime distribution of the dopaminergic subtypes presenting at least 50 cells, as defined in our annotation, for all the existing differentiation days or developmental timepoints.

We also computed the module activation scores (aggregated expression of log-normalized values) for each cell type to identify those modules whose expression was specific or enriched for a given cell type. We used the *aggregate_gene_expression*() function and the cell types defined in the unified annotation for that purpose. After selecting the five most highly activated modules in neurons (modules 6, 12, 18, 21 and 46), we performed gene ontology enrichment analysis to identify the potential biological processes driven by the genes in those modules. Using the *gost*() function from the *gprofiler2* R package again, we set the significance threshold at an adjusted P-value < 0.05 using the g:SCS method for multiple-test correction. We defined the gene universe in the "custom_bg" and used the functional annotation from the Cellular Component category of Gene Ontology (GO:CC)[88]. Based on the term size, the difference between the effective domain size and the term size, the query size and the intersection size, we computed the odds ratio for each term.

Finally, we also computed the module activation scores for each cell to compare the expression profile of the neuron-enriched modules

in the neurons profiled in our data versus the neurons from other datasets (unified annotation). Those were the comparisons:

This work (Fetal) vs Birtele et al. 2022 (Fetal)
This work (Fetal) vs Braun et al. 2023 (Fetal)

This work (Fetal) vs La Manno et al. 2016 (Fetal)
This work (in vitro) vs This work (Fetal)
This work (in vitro) vs Fiorenzano et al. 2021 (3D)

Given the differences in neuron cell number between the datasets, we performed a permutation test (*oneway_test*() function) resampling from the larger group in each comparison to match the smaller group (n = 100 times, mean p-value reported). For each comparison, we set the significance threshold at an adjusted P-value < 0.05 using the Benjamini & Hochberg multiple-test correction. The significance levels are indicated as follows: pAdj<0.001 (***), pAdj<0.01 (**) or pAdj<0.05 (*). Additionally, to capture the dynamics of gene module expression in neurons throughout midbrain development, as well to detect batch differences between datasets, we calculated the neuron-enriched module activation scores (mean ± standard deviation) for each of the developmental timepoints or differentiation days per dataset.

## Spatial transcriptomics
Snap frozen, OCT embedded tissue blocks were cryosectioned on a Bright OTF5000 cryostat. 10um sections were loaded onto Visium Spatial Gene Expression slides (product code: 1000184, 10X Genomics, Pleasanton, CA) as per 10X Genomics demonstrated protocol: CG000240. Triplicate sections of each of four gestational ages (7, 11, 12 and 17 PCWs) of brain tissue were randomized across the top three capture areas of 4 slides to avoid any potential batch effect. The fourth capture area of each slide was loaded with sections of OCT blocks, each containing six organoids randomized by donor, time point, differentiation and duplicate. Loaded Visium slides were subsequently methanol fixed, stained with Haematoxylin and Eosin (H&E) and imaged as per 10X Genomics demonstrated protocol: GC000160 on a Motic EasyScanOne slide scanner, at 40X magnification. Following 10X Genomics user guide: CG000239, sections were then subjected to permeabilization for mRNA capture, reverse transcription and second strand synthesis to generate full length cDNA, followed by library preparation. The permeabilization time of 12 minutes for both brain tissue and organoids was established as appropriate using the Visium Spatial Tissue Optimisation kit (product code: 1000193, 10X Genomics, Pleasanton, CA) following 10X Genomics demonstrated protocols and user guide: CG000240 for sectioning and slide loading, GC000160 for H&E staining and brightfield imaging and CG000238 for mRNA capture, reverse transcription, second strand synthesis and fluorescent imaging. The resulting libraries were sequenced on an Illumina Novaseq 6000 sequencer, using a Novaseq 6000 S2 (100 cycle) v2 sequencing kit.

## Spatial transcriptomics computational analysis
We used spaceranger (v2.1.0)[54] from 10X Genomics to process the bcl files obtained from the sequencer following spatial transcriptomics (ST) sequencing. We generated FASTQ files and gene expression matrices with the mkfastq and count commands with the ENSEMBL homo sapiens reference genome build GRCh38. For each sample, the FASTQ files, gene expression matrix and spaceranger output directory are available at the Gene Expression Omnibus (see data availability).

We conducted the subsequent analysis in Seurat (v4.4.0)[92], writing the code as an Rmarkdown notebook. All analysis was processed within Docker. The analysis can be replicated on any computer where Docker is available by downloading the data, code, and Docker images (see code availability).

We removed the 12 PCW sections from the dataset as the tissue had been damaged during processing. We also removed spots that were disconnected from the main slice of tissue/organoid. We identified progenitor and post-mitotic cell populations by clustering the data using Seurat's FindClusters function with sample-specific resolutions that generated two clusters. These two clusters were then annotated using marker genes and anatomical knowledge. When visualizing RNA molecules, we plotted, for each spot, a dot for each molecule detected (with multiple dots if more than one was detected there). We then slightly perturbed the positions of the dots to allow them all to be seen while keeping them near their associated spot.

Since each spot is likely to contain multiple cells (possibly of different cell types), we ran cell type deconvolution using Cell2Location (v0.1.3)[93] in conjunction with the annotated single cell reference that we developed in this paper.

To identify genes that are differentially expressed with respect to maturation, we first manually annotated the path of maturation on each sample and then used this as a covariate in CSIDE (from spacexr package v2.2.1)[54]. We provided CSIDE with our cell type deconvolutions, which allowed us to identify maturation-related differentially expressed genes for each cell type. We calculated the running weighted average of several genes of interest by averaging their expression along the path of maturation using a window size of 50 spots.

We profiled cell-cell interaction using COMMOT (v0.0.3)[66], which identifies the most-active spatially co-located ligand-receptor pairs, where the pairs were taken from the CellChat database (v1)[94]. We used Moran's I[95] to identify the ligand-receptor pairs with the highest spatial autocorrelation (i.e. where their position appears to explain their expression).

To quantify the spatial similarity between ST samples, we first labelled each spot as the cell type for which the deconvolution estimates it to have in the greatest abundance. We then counted the number of neighboring spots for each pair of cell types. We normalized these counts and used them to compute (Euclidean) distances between samples. Inverting and normalizing these distances to the range [0, 1] allowed us to treat them as percentages of similarity.

We estimated the temporal alignment of each organoid by computing a weighted average of the timepoints of the tissue samples, where weights were taken with respect to the organoid's spatial similarity to each timepoint.

## Immunoblotting

Proteins were extracted from cells in ice-cold RIPA lysis and extraction buffer (Sigma-Aldrich) supplemented with protease inhibitor[36]. Protein concentration was measured with Pierce™ BCA Protein Assay kit (Thermo Scientific): 10 μg of protein was denatured with Laemmli buffer (Bio-Rad Laboratories LTD) with dithiothreitol (DTT). Proteins were separated with Mini-PROTEAN TGX Stain Free Gels (Bio-Rad Laboratories LTD) and transferred to a Trans-Blot Turbo Transfer membrane (Bio-Rad Laboratories LTD). After blocking in 5% milk, 1x PBS, 0.1% Tween for 1 h at room temperature, membranes were incubated sequentially with primary antibodies at 4 °C overnight, with antibodies removal performed using Restore Western Blot Stripping Buffer (Thermo Scientific) between incubations (Supp Data S12). Membranes were then incubated with the secondary anti-rabbit horseradish peroxidase-conjugated antibody at a dilution of 1:3000 (Cell Signalling). Immunoreactive proteins were visualized with Chemidoc MP (Bio-Rad Laboratories). Equal loading was evaluated with probing for GAPDH, membranes were cut at -50 kDa to permit co-blotting of the protein of interest and GAPDH. The intensity of immunoreactive bands was analyzed using ImageJ software (National Institutes of Health). The density of the bands was normalized to GAPDH. Results are reported as means SEM of independent experiments, the number of which is stated for each experiment in the respective figure legend.

## Statistics and reproducibility

Throughout the manuscript, 2D and 3D MLO micrographs were consistently acquired from a minimum of three independent differentiations. Only representative images are presented in the manuscript. For fetal midbrain samples, a minimum of one micrograph per sample, taken across at least one rostro-caudal section, was obtained due to the limited availability of these specimens. For the statistical analysis of iPSCs derived data, two-tailed Student's t-test was applied for dual analysis comparison. Results are reported as mean ± standard error or the mean (SEM) from at least three independent biological replicates. Significant levels were determined by P-value. P-values are reported as exact values in all figures. Analysis was performed using GraphPad Prism.

## Ethical approval

The UK NHS Health Research Authority, London Hampstead Research Ethics Committee, has approved the work with induced pluripotent stem cells under the REC reference number 13/LO/0171, IRAS project ID: 100318.

Fetal samples have been collected under the ethical approval REC reference number 23/NE/0135, IRAS project ID: 330783 and REC reference number 23/LO/0312, IRAS project ID: 326492, granted to Human Developmental Biology Resource (HDBR) by UK NHS Health Research Authority, North East - Newcastle & North Tyneside 1 Research Ethics Committee.

## Reporting summary

Further information on research design is available in the Nature Portfolio Reporting Summary linked to this article.

## Data availability

Raw FASTQ files for both single-cell and spatial transcriptomics generated in this study have been deposited in Gene Expression Omnibus (GEO) under accession code GSE277032, accessible at https://www.ncbi.nlm.nih.gov/geo/query/acc.cgi?acc=GSE277032. The four supplemental data files generated in this study are available in a Zenodo repository (https://zenodo.org/records/13765496), detailed as follows: Files S1, S2: VCF files with donor genotypes to deconvolute pooled scRNA-seq data. File S3: Single-cell data generated in our study (RDS object: Seurat object with counts and metadata). File S4: Integrated datasets (RDS object: Seurat object with counts and metadata). Additionally, the images from the immunohistochemistry analysis on midbrain tissues are available in the MRC-Wellcome Trust Human Developmental Biology Resource (HDBR) Atlas (https://hdbratlas.org/gene-expression/midbrain.html). Source data for graphs and tables are included with the manuscript. Source data are provided with this paper.

## Code availability

Two GitHub repositories are available for reproducing the analysis and figures of the manuscript: one for single-cell transcriptomics (https://github.com/paupuigdevall/MLOscRNAseq, archived at https://doi.org/10.5281/zenodo.17541380) and another for spatial transcriptomics (https://github.com/george-hall-ucl/mlo_spatial_transcriptomics, archived at https://doi.org/10.5281/zenodo.17550243). These repositories include scripts for processing steps, data analysis and figure generation. Additionally, the Docker images used for spatial transcriptomics are available on Dockerhub (main pipeline: georgehallucl/midbrain_tissues_organoids_docker; commot: georgehallucl/commot_docker; cell2location: georgehallucl/cell2location_docker). An archive to reproduce the spatial transcriptomics analysis is available at https://doi.org/10.5522/04/27044423.

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

## Acknowledgements

We sincerely thank our patients and their families for participating in this study. D.B. was supported by the Sir Jules Thorn Trust. P.P.C. and H.K. are currently funded by the Sigrid Jusélius Foundation and previously by the NIHR Great Ormond Street Hospital Biomedical Research Centre. G.T.H. is funded by the NIHR Great Ormond Street Hospital Biomedical Research Centre. J.J. was supported by a postdoctoral fellowship from OpenTargets. F.P. is supported by the UK Medical Research Council (MRC) [UKRI222 – PI S.B.]. M.A.K. is supported by the NIHR Professorship, the Sir Jules Thorn Award for Biomedical Research and the Rosetrees Trust. S.B. was supported by the Great Ormond Street Hospital Children's Charity. We thank UCL Genomics (UCL GOS Institute of Child Health) and the Wellcome Sanger Institute for undertaking scRNA and spatial sequencing. We thank the Human Developmental Biology Resource (HDBR) for providing human midbrain fetal tissue. We thank the UCL Imaging Facility and the facility manager, Dr Dale Moulding, for the support with immunofluorescence analysis. This research was supported by the NIHR Great Ormond Street Hospital Biomedical Research Centre. The views expressed are those of the author(s) and not necessarily those of the NHS, the NIHR or the Department of Health. Graphical schemes have been created in BioRender. https://BioRender.com/.

## Author contributions

D.B., S.B., and M.A.K. conceived and designed the study. S.B. designed and performed experiments and data analysis for the in vitro and in vivo study. D.B. designed and performed experiments and data analysis for the in vitro and in vivo study. P.P.C. designed and performed data analysis for the scRNA-seq study. G.T.H. designed and performed data analysis for the spatial transcriptomic study. C.R. performed experiments for the in vitro study. T.X. designed and performed experiments for the scRNA-seq and spatial transcriptomic studies. E.M. performed experiments for the spatial transcriptomics study. J.J. performed experiments for the scRNA-seq study. A.D.D. performed experiments for the in vitro study. F.P. performed experiments for the in vivo study. H.K. supervised the scRNA-seq study. S.C. supervised scRNA-seq and spatial transcriptomic studies. D.B., P.P.C., G.T.H., S.C., S.B., and M.A.K. drafted the manuscript. C.R. and T.X. contributed to the written sections of the manuscript. All authors reviewed the manuscript prior to submission.

## Competing interests

M.A.K. is a founder of and consultant to Bloomsbury Genetic Therapies. She has received honoraria from PTC for sponsored symposia and provided consultancy. The remaining authors declare no competing interests.
