## [Transparent Peer Review file · Nature Communications]

An in vivo and in vitro spatiotemporal profile of human midbrain development

Corresponding Author: Dr Serena Barral

Version 0:

Reviewer comments:

Reviewer #1

(Remarks to the Author)

Studying the unique features of human brain development in vivo is hampered by the ethical and practical issues concerning the use of fetal tissue. As an alternative, in vitro 2D or 3D (organoid) models either from iPSC cells or hESC lines have been developed to mimic the early stages of nervous system formation. The main open question of the field is, how reliable are these models – i.e. how well they can model the in vivo brain development.

In this work the authors have addressed this question by an extensive analysis of fetal human midbrain, iPSC-derived midbrain organoids, and 2D cell cultures which have been directed to adopt midbrain fate, at a single cell resolution. Compared to a recent work by Fiorenzano et al (Nat. Comm 2021), which also analysed midbrain organoids by scRNAseq and compared the results to previously published datasets of fetal midbrain of first trimester, this work includes also tissue from second trimester and the authors have done scRNAseq themselves, including spatial transcriptomics. They have compared their results multiple previous datasets of midbrain organoids, postmortem tissue, and fetal midbrain. Finally, they use a patient-derived midbrain organoid system to model DTDS.

The project is very extensive, and would potentially be highly valuable for the scientific community. Unfortunately, the manuscript is hard to read due to problems with data presentation, and the execution leaves a lot to be desired – so much so that it is difficult to evaluate whether central of the authors' claims are true or not.

Major points

1. The majority of immunostainings are not of good enough quality to judge whether certain cells and/or tissues do express a particular gene or not. Many transcription factors, which should show a clear nuclear signal, appear cytoplasmic instead – for example Lmx1 in Fig2C, G; Supp Fig 5; Sox2 in Fig5D, F6C. Sometimes the signal is so weak that it is unclear if it is true or only background – e.g. FoxA2 in Supp Fig 1B, TH in Supp Fig 2B where it is supposed to mark SN but the signal cannot be seen; GABA in Supp Fig 3A and Corin in Fig 2 and Supp Fig4. As the both fetal and organoid sections also appear torn, containing big holes, or otherwise damaged, it suggests a technical problem in tissue fixation, sectioning or other processing. According to Methods, the authors have fixed organoids only 30 minutes and human fetal midbrains overnight in ice cold 4% PFA – these times are clearly not enough. For an average size organoid a fixation time of several hours is more optimal (eg. Methods in Fiorenzano et al., 2021). The organoid and also some cell culture IHC stainings should be repeated with optimized conditions for fixation, sectioning, and IHC. It is unclear how the sections were imaged as there is no information about this in Methods (?!), but confocal settings should also be adjusted for improved image acquisition.

2. Statements like "...we observed fewer Ki67+ cells" (line 160) must be accompanied by quantification.

The manuscript is very hard to read due to problems in data presentation, for example:

3. Texts and details in many figures are too small to read – for example the text in the dot plot in Supp Fig 7A is so small that many gene names were unreadable even when zooming in on a high resolution display (turning the plot direction vertically and making it the length of the entire page should help). Same applies to individual images from spatial transcriptomics which are very hard to see e.g. in Supp Fig 10.

4. Many labels within the figures are hard to decipher and one wonders if there could be clearer labels for data than e.g.

"organoid_names_OCT_BLOCK_1 OCT_BLOCK1" in Supp Fig 9C (another image where there are apparently some labels within the images but they are unreadable); "PCW_17_REP_3", or "Donor1_day40_dif_org2" (Fig. 6E,F).

5. Sometimes critical labels are missing entirely making the figures impossible to understand – for example Supp Fig 17A and Supp Fig 18A should, according to the figure legends, contain IHC images of organoids from Patients 1, 2, Controls 1,2 and isogenic control, but the reader is left guessing which is which.

6. UMAP coordinates in Fig 4E are not fixed which results in "elongated" UMAPs compared to Fig 4A, C, D – making it harder to compare these to each other.

7. In Supp Fig 8H plot the colour code for both data from this work and previous dataset are both labelled white.

8. The colour codes for scRNAseq data are apparently the default ones which makes some of the colours to be so similar to each other that they cannot be separated by eye even by zooming in on a display. For example in Fig3D, hDA3 and hEndo are almost same shade of green, an hRGl2 and hRGl3 are almost same shade of brown. These colour codes should be customized in R so that one can easily tell them apart.

9. In Figure 5G, the colourcode for each line is nearly unreadable – "PCW_7_REP3" and "Donor1_day40_dif_org2" look the same, for example.

10. Scalebars are often absent and some images which, based on one scale bar in one of the figures should be taken on a same magnification, look very different. For example Supp Fig 4F appears to be in a higher magnification than Supp Fig 4G, based on the size of the nuclei. Yet only scale bar is presented in Supp Fig 4H, which also seems to be a different magnification compared to the other two.

11. The labels for the stainings are placed within the images making sometimes very hard to read – they would be better placed outside.

12. For their scRNAseq analyses, the authors have decided to use labels/nomenclature from previous datasets (La Manno et al., 2016 and Fiorenzano et al., 2021). This results in overly complicated datasets with unnecessarily numerous subgroups. This is especially problematic as many subgroup identities from La Manno et al are unclear in a developmental biology context – they for example talk about progenitors and radial glia as separate types although radial glia *are* progenitors. In addition, the term "neuroblast" originates from a Drosophila and should not be used for mammalian cells as it does not have an exact counterpart there (although one unfortunately sees it a lot in that context). The datasets in this work would have been clearer if they had identified the clusters in a more unbiased way, dividing them broadly into, for example, mDA progenitors, non-mDA progenitors, immature mDA and non-mDA precursors, mature neurons (with possible mDA subtypes resolved e.g. by hierarchical clustering), and different non-neuronal types as their own groups.

Minor point

13. The authors report that scRNAseq of the fetal midbrain yielded enrichment of mixed hDA3 population and depletion of other cell types. This might be a technical issue – whole mDA neurons can be difficult to isolate from the tissue. Was there a reason why single nuclei RNA sequencing – which gives better yields compared to whole cell RNAseq - was not applied instead?

(Remarks on code availability)

Reviewer #2

(Remarks to the Author)

In the manuscript "An in vivo and in vitro spatiotemporal atlas of human midbrain development" the authors profiled first and second trimester fetal midbrain tissue to compare it with in vitro models derived from pluripotent stem cells. They integrated atlases and found that in vitro models recapitulate the transcriptional activity of late first trimester fetal midbrain, while 3D models replicate the spatial organization and cellular architecture of first and second trimester midbrain. This is an important study that highlights the strengths and limitations of lab models for studying dopaminergic system development.

The study carried out by the authors is compelling, and their effort to integrate single-cell datasets presented in the study with existing ones is commendable. The profiling of fetal tissue 6-22 post conceptional weeks is a unique resource and forms a novel dataset well worth publishing. Moreover, the findings related to DTDS are certainly intriguing and provides a valuable example of how the resulting atlas can be leveraged to analyze developmental trajectories, maturation, and the diversity of both patient-derived and control midbrain organoids.

However, there are some weaknesses in the study that need to be addressed prior to publication. This study uses advanced technology but the data obtained appeared not explored enough, even the spatial transcriptomics is definitely an interesting analysis but in the end it appears do not add much at the meaning of manuscript. On their own, these data add little to the overall understanding of dopaminergic neuron development. It would be far more impactful if the authors explored the

developmental layering of neurons and the factors driving the spatial segregation of dopaminergic subtypes. Overall, this work has significant potential, but it seems that the valuable resource the authors have created has not been fully leveraged:

Major revisions:

- The histological analysis of embryos 13, 15, 19 and 22 PCW is an important resource but needs extension to be fully informative. For example, expression of A9 vs A10 specific proteins ALDHA1A, PITX3, OTX2, GIRK2, CALB in the TH+ neurons should be analysed.
- Profiling a total of 34,191 cells from six donors, pooled into 13 samples (with 10X libraries prepared from a single channel), represents a relatively low number of profiled cells for a study aiming to create an in vivo and in vitro spatiotemporal atlas of human midbrain development. While we understand the challenges of working with precious fetal material, particularly from the second trimester—a real novel data source in this study—the limited number of dopaminergic neurons captured at single cell resolution emerging from these datasets limits the impact of the findings. Increasing the number of profiled cells (in particular DA neurons) would be necessary for robustness and overall impact of the findings. Maybe this can be achieved via nuclei seq?
- The presentation of different clusters (i.e. Fig 3B-D) gets very hard to digest with this many celltypes. Especillay 3C is almost impossible to decode. Would the clarity be improved by splitting the cells into major cell types (neurons, glia, progenitors, other) and then perform characterization/visualisations per major cell type?
- The developmental trajectories of dopamine (DA) neurons and non-neuronal cell types in the fetal samples should be traced so that the analysis not merely complement existing human datasets on midbrain development (as mentioned in line 512) but stands as a valuable reference in its own right.
- The authors state that “generally, neuron proportions decreased at later stages of fetal development, with a simultaneous increase in glial cells, microglia, and oligodendrocyte cell lineages. Unsurprisingly, the proportion of neurons in adult post-mortem samples was only 3.3%.” While this is understandable, given that fetal tissue contains a greater variety of cell types compared to in vitro hPSC cultures, the authors should consider the possibility that tissue dissociation during single-cell preparation may disproportionately affect neurons compared to other cell types. Additionally, Figure 3D clearly shows that samples from PCW16 and PCW20—representing the real novel data in this manuscript—contain very few DA neurons, which warrants further consideration. Using scRNA-seq rather than nuclei makes one wonder which fragile cell types (i.e. neurons) are missed?
- The spatial transcriptomics used in this study has a low resolution, which limits its ability to provide detailed biological insights. As shown in Figure 5A–E, the resolution (~55 µm per spot) results in broad clustering of cell types rather than precise localization. For example, the spatial distribution of progenitor and post-mitotic cells lacks the granularity needed to separate closely packed cell populations or specific niches. While this method gives a general overview of cellular organization, it should be complemented with supporting immunostainings.

Minor points

- The exact number of cells per replicate needs to be clearly stated in the main text or figures.
- Why not use the established acronym VLMC instead of “vascLepto”?
- 6D, legend?
- In fig 7, does (C) refer to d40 and (D) to d70?
- For the data in Fig 7, where there any cell type abundance differences between patients and controls?

(Remarks on code availability)

Reviewer #4

(Remarks to the Author)

(Remarks on code availability)

Yes, I have assessed the code, and I find it to be a well-prepared and valuable resource for the community. The results of the paper are reproducible to a significant extent, as the code is clearly structured and provides all the necessary components to replicate the experiments described.

Reviewer #6

(Remarks to the Author)

(Remarks on code availability)

Version 1:

Reviewer comments:

Reviewer #1

(Remarks to the Author)

No further comments.

(Remarks on code availability)

Reviewer #2

(Remarks to the Author)

The authors have made substantial and important revisions that greatly improved the manuscript. We're satisfied with the revisions and recommend publication

(Remarks on code availability)

Reviewer #4

(Remarks to the Author)

(Remarks on code availability)

Reviewer #6

(Remarks to the Author)

(Remarks on code availability)

Reviewers comments

Reviewer #1

Remarks to the Author:

Studying the unique features of human brain development in vivo is hampered by the ethical and practical issues concerning the use of fetal tissue. As an alternative, in vitro 2D or 3D (organoid) models either from iPSC cells or hESC lines have been developed to mimic the early stages of nervous system formation. The main open question of the field is, how reliable are these models – i.e. how well they can model the in vivo brain development.

In this work the authors have addressed this question by an extensive analysis of fetal human midbrain, iPSC-derived midbrain organoids, and 2D cell cultures which have been directed to adopt midbrain fate, at a single cell resolution. Compared to a recent work by Fiorenzano et al (Nat. Comm 2021), which also analysed midbrain organoids by scRNAseq and compared the results to previously published datasets of fetal midbrain of first trimester, this work includes also tissue from second trimester and the authors have done scRNAseq themselves, including spatial transcriptomics. They have compared their results multiple previous datasets of midbrain organoids, postmortem tissue, and fetal midbrain. Finally, they use a patient-derived midbrain organoid system to model DTDS.

The project is very extensive and would potentially be highly valuable for the scientific community. Unfortunately, the manuscript is hard to read due to problems with data presentation, and the execution leaves a lot to be desired – so much so that it is difficult to evaluate whether central of the authors' claims are true or not.

Author response:

We would like to thank the reviewer for their detailed and insightful feedback on our manuscript. We appreciate their recognition of the potential value of our extensive analysis for the scientific community.

In response to the reviewer's concerns about data presentation, we have made several revisions to improve the clarity and readability of our manuscript:

- We have revised our figures and tables to present the data more clearly and concisely. This includes the use of more intuitive color schemes, improved labeling, and explanatory legends. Figure changes have been summarized in a document attached to this rebuttal.
- We have restructured the “Results” and “Discussion” sections to more clearly highlight the key findings and their implications. We have also included additional context to better explain how our results compare to previous studies.

- We have carefully considered the reviewer's comments and have addressed below the specific questions regarding the execution of our experiments. This includes providing additional analysis and data where necessary and clarifying any ambiguities in our original submission. We hope that these revisions address all concerns and enhance the overall quality of our manuscript.

Major points:

1. The majority of immunostainings are not of good enough quality to judge whether certain cells and/or tissues do express a particular gene or not. Many transcription factors, which should show a clear nuclear signal, appear cytoplasmic instead – for example Lmx1 in Fig2C, G; Supp Fig 5; Sox2 in Fig5D, F6C. Sometimes the signal is so weak that it is unclear if it is true or only background – e.g. FoxA2 in Supp Fig 1B, TH in Supp Fig 2B where it is supposed to mark SN but the signal cannot be seen; GABA in Supp Fig 3A and Corin in Fig 2 and Supp Fig4. As both the fetal and organoid sections also appear torn, containing big holes, or otherwise damaged, it suggests a technical problem in tissue fixation, sectioning or other processing. According to Methods, the authors have fixed organoids only 30 minutes and human fetal midbrains overnight in ice cold 4% PFA – these times are clearly not enough. For an average size organoid a fixation time of several hours is more optimal (eg. Methods in Fiorenzano et al., 2021). The organoid and also some cell culture IHC stainings should be repeated with optimized conditions for fixation, sectioning, and IHC. It is unclear how the sections were imaged as there is no information about this in Methods (?!), but confocal settings should also be adjusted for improved image acquisition.

Author response:

We thank the reviewer for the in-depth evaluation of our immunofluorescence analysis and for the opportunity to improve the quality of our work.

We believe that the overall quality of the immunostainings was compromised when we embedded the figures as TIFF files into the Word document for submission to the journal. We apologize for the inconvenience. This issue has been resolved in the revised rebuttal document. Additionally, we have revised all our images, conducted new analysis and improved image contrast as recommended by the reviewer. We summarize the changes made below. A detailed list of all images modified in each figure can be found in TrackFigureChange excel document submitted with this rebuttal.

- We agree that immunofluorescence analysis using the LMX1 antibody AB10533 (Millipore) shows cytoplasmic localization in some images. This antibody has already been reported to show non-specific staining in low-density cultures by the Parmar group (PMID: 28858290). To avoid any unclear signal, we have tested another antibody (LMX1A - Abcam, ab139726) and performed a new immunofluorescence analysis for LMX1A across our samples (see table below for details on each image changed).

- The SOX2 antibodies used in this study (antibodies SOX2 - Cell Signaling Technology and SOX2 - R&D Systems MAB2018) demonstrate clear specificity. The SOX2 staining shown in Figures 5D and 6C is indeed nuclear (**new Figures 7C and Supplementary Figures 14-15**). However, we acknowledge that the tissue presented in the spatial section of our study may not offer optimal conditions for immunohistochemistry (IHC) analysis. The sections shown are adjacent to those used for the spatial transcriptomics analysis. These tissues were snap-frozen, sectioned under cold conditions, and either directly processed for spatial analysis or post-fixed in 2% PFA and processed for immunofluorescence (as similarly shown in PMID: 33949649). To provide better IHC analysis, we have collected additional fetal samples at 7, 11, and 17 PCW, processed them under optimized IHC conditions, and repeated the analysis (**Supplementary Figure 14**).

- We have revised all images where the signal was too weak, hindering proper evaluation of our analysis. Specifically:
 - o **FoxA2 in Supplementary Figure 1B:** We performed new immunostaining on a 6 PCW fetal sample.
 - o **TH in Supplementary Figure 2B:** We replaced the human midbrain samples at 15 PCW with a sample at 16 PCW and improved resolution. We also provide external access to high-resolution images from the second trimester fetal samples featured in the manuscript figures. In line with this, all our immunofluorescence images will be made available on the HDBR webpage (Methods section – page 26, Lines 752-758).
 - o **GABA in Supplementary Figure 5A and Corin in Figure 2 and Supplementary Figure 6:** We improved image quality and enhanced the contrast of the immunofluorescence dataset.

- We apologize for the lack of clarity in the “Immunostaining” subsection of the Methods (**Page 27, lines 786-793**). We have now revised this section to include the actual fixation timing for each sample type, details of the imaging equipment and acquisition settings used, and the image analysis procedures performed. To improve the quality of the data, we have performed new sectioning and immunofluorescence on both previously collected and newly derived organoids. In addition, we have replaced fetal samples that showed significant tissue damage with newly collected samples at 12 and 16 PCW (**Figure 1 and Supplementary figures 2-4**). As previously mentioned, we have also collected new fetal samples at 7, 11, and 17 PCW, which were specifically processed for IHC. These new samples have been used to update the immunofluorescence imaging in current **Figure 7 and Supplementary Figure 14** (see the attached summary table for details). Based on our prior experience with forebrain assembloids and brain tissue from animal models (PMIDs: 36378958, 34011628, 28943253, 27767083, 23237958), we are confident that our fixation conditions are appropriate and do not lead to under-fixation and avoid over-fixation of the tissue, which could compromise specificity in immunofluorescence detection. Our 3D midbrain organoids are embedded in Matrigel, which supports cellular reorganization and preserves overall organoid architecture. As a result, midbrain organoids commonly have

internal cavities or neurite arborization extending into residual Matrigel. This phenomenon is also observed in Fiorenzano et al., 2021 (PMID: 34911939, Figures 1l–1n), as well as in other published studies (PMID: 34027482, 36121202). We have now clearly indicated when imaging was performed in Matrigel-rich regions (indicated has Matrigel embedding [me])—for example, in our GIRK2 analysis, where neurite distribution is more evident within the Matrigel extension. Nonetheless, we agree that the initial set of images displayed signs of breakage in the necrotic core of some midbrain organoids, potentially due to use of an outdated cryostat. To address this, we have repeated the analysis and updated the data analysis (see attached table TrackFigureChange). Addressing the reviewer comment on fetal tissue sections, we collected additional samples to replicate the IHC analysis and ensure that results were reproducible despite the challenges of sample collection (surgical termination, dissections, overnight shipment).

2. Statements like “...we observed fewer Ki67+ cells” (line 160) must be accompanied by quantification.

Author response:

We agree with the reviewer about the lack of quantification. We now provide a Ki67 quantification in support of our statement, which can be seen in Figure S8A.

3. Texts and details in many figures are too small to read – for example the text in the dot plot in Supp Fig 7A is so small that many gene names were unreadable even when zooming in on a high resolution display (turning the plot direction vertically and making it the length of the entire page should help). Same applies to individual images from spatial transcriptomics which are very hard to see e.g. in Supp Fig 10.

Author response:

We thank the reviewer for alerting us about the small font size of both panels of the supplemental figure. As suggested, we have rotated Supp Fig 9A vertically and increased its resolution for improved readability. In addition, we have created a new panel (Supp Fig 9B), which expands on old Figure 3B by including the module activation score distribution for all cell-type specific markers shown in Supp Fig 9A. The remaining panels from the original Supp Fig 7 (B-F) have been moved to the new Supp Fig 10 (B-F), while the space previously occupied by Supp Fig 7A is now used for a new panel, Supp Fig 10A. This panel provides an overview of the dataset structure for the in vitro and fetal samples – including donor, replicate and 10x library information – as requested by Reviewer 2. As for the spatial transcriptomic figures, we have increased the sizes of images and text to improve readability. For example, in figures where the expression of two markers was plotted side-by-side (e.g. Fig 5C, Supp. Fig 10), we have now combined these marker plots into a single one, thereby increasing resolution (Figure 5C, Supp. Fig 16).

4. Many labels within the figures are hard to decipher and one wonders if there could be clearer labels for data than e.g. “organoid_names_OCT_BLOCK_1 OCT_BLOCK1” in Supp Fig 9C (another image where there are apparently some labels within the images but they are unreadable); “PCW_17_REP_3”, or “Donor1_day40_dif_org2” (Fig. 6E,F).

Author response:

Thank you for pointing out that these labels were difficult to read. We have now improved their readability (Supp Fig 15).

5. Sometimes critical labels are missing entirely making the figures impossible to understand – for example Supp Fig 17A and Supp Fig 18A should, according to the figure legends, contain IHC images of organoids from Patients 1, 2, Controls 1,2 and isogenic control, but the reader is left guessing which is which.

Author response:

We appreciate the reviewer’s suggestion to add a figure label for clearer identification of the samples. We have now incorporated this change in these figures, which are now new Supp Figures 23,24.

6. UMAP coordinates in Fig 4E are not fixed which results in “elongated” UMAPs compared to Fig 4A, C, D – making it harder to compare these to each other.

Author response:

We have revised Fig 4E to ensure that the multi-faceted UMAPs maintain the same height-to-width ratio as the other panels (Fig 4A, 4C, 4D), avoiding distortion of the UMAP coordinates and making it easier to compare to the others.

7. In Supp Fig 8H plot the color code for both data from this work and previous dataset are both labelled white.

Author response:

A higher-resolution version of the original Supp Fig 8H panel (now Supp Figure 13D) was generated with clearly visible and distinct fill colors. We apologize for not detecting this issue earlier, which was due to document compression at the time of submission.

8. The colour codes for scRNAseq data are apparently the default ones which makes some of the colours to be so similar to each other that they cannot be separated by eye even by zooming in on a display. For example in Fig3D, hDA3 and hEndo are almost same shade of green, an hRG12 and hRG13 are almost same shade of brown. These colour codes should be customized in R so that one can easily tell them apart.

Author response:

We appreciate the reviewer's point regarding the very reasonable confusion caused by using a 24-color palette to visualize all cell types simultaneously in UMAPs and stacked bar plots. Although we made efforts to customize the palette, some color combinations remain difficult to distinguish between these many cell types. To address this – also in line with major point 12 below – we now present a low-resolution annotation (10 broader groups) in main Figure 3 (panels C and D), improving clarity in both UMAPs and compositional barplots. Details on how this low-resolution annotation was generated are provided in our response to point 12, as well as in Methods (Pages 31-32, lines 907-923).

The original 24-cell type annotation (high-resolution) and its full color palette have been retained in a single supplementary figure where color discrimination is not a concern: the faceted UMAP in Supp Figure 10B (formerly Supp Figure 7B). This will allow the reader to still access this original annotation, which remains a useful reference and is consistent with previous efforts in the midbrain field (PMID: 27716510).

9. In Figure 5G, the colour code for each line is nearly unreadable – “PCW_7_REP3” and “Donor1_day40_dif_org2” look the same, for example.

Author response:

We thank the reviewer for pointing this out. We have now revised Figure 5E (old figure 5G) with more distinct color codes for better clarity and have increased the width of the lines in the legend.

10. Scalebars are often absent and some images which, based on one scale bar in one of the figures should be taken on a same magnification, look very different. For example Supp Fig 4F appears to be in a higher magnification than Supp Fig 4G, based on the size of the nuclei. Yet only scale bar is presented in Supp Fig 4H, which also seems to be a different magnification compared to the other two.

Author response:

We thank the reviewer for the thorough analysis of our immunofluorescence images. When using the same scale bar across a series of images, we confirm that all were acquired at the same magnification and processed under identical conditions. We have increased the DAPI intensity to enhance visualization of the nuclei.

Regarding the specific images highlighted by the reviewer—Supplementary Figure 4G in comparison to Figures 4F and 4H—we believe the observed differences may reflect an increase in the size of KI67-positive nuclei. Nuclear size can vary throughout the cell cycle, and KI67 is expressed across the G2–M phases, during which notable nuclear changes occur (PMID: 21738834; PMID: 28900009). Therefore, we cannot exclude the possibility that this is the effect observed by the reviewer.

Additionally, differences in nuclear appearance may also result from variations in cell density within the imaged area, slight differences in imaging settings for each marker, or post-acquisition image adjustments.

11. The labels for the stainings are placed within the images making sometimes very hard to read – they would be better placed outside.

Author response:

We agree with the reviewer’s comment regarding the immunostaining labels and have now placed them outside the images (when space allows) throughout the whole manuscript for better clarity.

12. For their scRNAseq analyses, the authors have decided to use labels/nomenclature from previous datasets (La Manno et al., 2016 and Fiorenzano et al., 2021). This results in overly complicated datasets with unnecessarily numerous subgroups. This is especially problematic as many subgroup identities from La Manno et al are unclear in a developmental biology context – they for example talk about progenitors and radial glia as separate types although radial glia *are* progenitors. In addition, the term “neuroblast” originates from a Drosophila and should not be used for mammalian cells as it does not have an exact counterpart there (although one unfortunately sees it a lot in that context). The datasets in this work would have been clearer if they had identified the clusters in a more unbiased way, dividing them broadly into, for example, mDA progenitors, non-mDA progenitors, immature mDA and non-mDA precursors, mature neurons (with possible mDA subtypes resolved e.g. by hierarchical clustering), and different non-neuronal types as their own groups.

Author response:

We recognize that the large number of subtypes initially presented in this study can make visualization challenging. Cell type annotation remains a complex task, both in terms of cell type identity and resolution. In this sense, the La Manno et al. annotation (PMID 27716510) provides a high-resolution reference that, unlike the present study, did not profile thousands of cells. Our initial classification, which partially aligns with La Manno’s, remains a valuable resource, as it is grounded in an extensive literature-based markers.

Nevertheless, as demonstrated by our multi-dataset integration, it is often necessary to provide a low-resolution annotation to enable meaningful cross-study comparisons (i.e. Birtele et al. 2022, Braun et al. 2023, Fiorenzano et al. 2021, Agarwal et al. 2020). In response to the reviewer’s comment – and in line with current practices in single-cell transcriptomics – we have now included a low-resolution annotation for our dataset. This annotation is presented in the main figures (Figure 3, Supp Figures 9-10) to improve clarity, while the initial high-resolution annotation is retained to capture finer differences in subtype composition and pseudotime trajectories.

To generate the new annotation, we initially performed hierarchical Leiden clustering using varying resolutions (res = 0.1, 0.3, 0.5, 1) in Seurat. Our original high-resolution annotation (24 clusters) was based on res=0.75. For the low-resolution annotation, we used the lowest resolution clustering (res=0.1, 9 clusters) as a guide to define 10 broader cell categories. This new classification represents a compromise that incorporates both reviewers' feedback while preserving biologically meaningful groupings. It reflects developmental trajectories by separating progenitor states, lineage-specific precursors and mature cell types. Additionally, non-neuronal populations (i.e. glia, microglia and perivascular) are kept as separate categories due to their distinct origins and functions.

The 10 new categories are listed below, with the corresponding high-resolution subtypes in parentheses. We also have updated Table S3 to incorporate the two annotations.

- **Astrocytes** (*Astro*)
- **Early Midbrain progenitors** (*hProgFPM, hMidPre, hRgl1, hRgl4/MultiEpend*)
- **Late Midbrain progenitors** (*OPC1, OPC2, hRgl2/immAstro, hRgl3_caudal*)
- **Highly proliferative NSC** (*hProgM*)
- **Neuron progenitors** (*hPreDA, hNPro*)
- **Immature neurons** (*hNbDA, hNbGaba*)
- **Mature neurons** (*hDA1a, hDA1b, hDA2, hDA3/hGABA/hSer*)
- **Microglia** (*hMgl*)
- **Perivascular cells** (*VLMC, hPeric, hEndo*)
- **Others** (*Unknown, erythrocytes*).

For consistency, we have also grouped progenitors and radial glia under the same group for both the new low-resolution annotation, and the unified annotation of the six integrated datasets (Table S6). Similarly, we now refer to immature neurons instead of neuroblasts, hence we have updated and simplified the annotation in the corresponding figures (see the attached document: “Tracking_Figure_Changes.xlsx”).

Minor point:

13. The authors report that scRNAseq of the fetal midbrain yielded enrichment of mixed hDA3 population and depletion of other cell types. This might be a technical issue – whole mDA neurons can be difficult to isolate from the tissue. Was there a reason why single nuclei RNA sequencing – which gives better yields compared to whole cell RNAseq - was not applied instead?

Author response:

Our decision to use scRNAseq was based on the following considerations:

- scRNAseq allows for the capture and analysis of RNA from the entire cell, providing a more comprehensive view of the transcriptome, including both nuclear and cytoplasmic RNA. This was particularly important for our study, as we aimed to capture the full

spectrum of gene expression to evaluate the similarities between in vitro human iPSC-derived midbrain models and the developing human midbrain. This has allowed us to assess the reliability of these models for studying midbrain neurological disorders—many of which involve mitochondrial dysfunction (PMID: 33298971). For example, we hypothesize that neurodegeneration in Dopamine Transporter Deficiency Syndrome (DTDS) may be linked to mitochondrial dysfunction, similar to what is observed in adult neurodegenerative disorders such as Parkinson's disease (PMID 37951933, 33074190). Our scRNA-seq approach enabled us to identify dysregulation of mitochondria-related genes specifically in dopaminergic neurons within the in vitro DTDS model (Fig. 8 and associated Supp Figures 19-24). For this reason, it was essential to generate a comprehensive scRNA-seq map that includes transcriptomic information beyond the nuclear compartment. Moreover, mitochondria play a critical role in neuronal maturation, and we observed transcriptomic differences between midbrain in vitro models that further underscore the importance of mitochondrial function (Fig 3G) Therefore, we believe that a scRNA-seq approach is particularly well-suited to address these specific objectives of our study.

- Our study aimed to compare our findings with fetal and in vitro models existing datasets, many of which were generated using scRNAseq (i.e. Birtele et al. 2022, Braun et al. 2023, Fiorenzano et al. 2021, Agarwal et al. 2020), rather than single nuclei RNA sequencing, which allowed for more direct and meaningful comparisons.
- We agree that mDA neurons can be technically challenging to isolate, and applied the same processing pipeline to all samples for accurate comparisons. Indeed, the depletion of mDA neurons can be representative of the increasing difficulty of isolating them due to developmental changes. However, we believe that we have isolated a sufficient number of DA neurons for downstream analyses.

We added a paragraph in the discussion section to reflect this choice and the limitations of using this method (Discussion section, page 23, lines 675-685).

Reviewer #2

Remarks to the Author:

In the manuscript "An in vivo and in vitro spatiotemporal atlas of human midbrain development" the authors profiled first and second trimester fetal midbrain tissue to compare it with in vitro models derived from pluripotent stem cells. They integrated atlases and found that in vitro models recapitulate the transcriptional activity of late first trimester fetal midbrain, while 3D models replicate the spatial organization and cellular architecture of first and second trimester midbrain. This is an important study that highlights the strengths and limitations of lab models for studying dopaminergic system development.

The study carried out by the authors is compelling, and their effort to integrate single-cell datasets presented in the study with existing ones is commendable. The profiling of fetal tissue 6-22 post conceptional weeks is a unique resource and forms a novel dataset well worth publishing. Moreover, the findings related to DTDS are certainly intriguing and provides a valuable example of how the resulting atlas can be leveraged to analyze developmental trajectories, maturation, and the diversity of both patient-derived and control midbrain organoids.

However, there are some weaknesses in the study that need to be addressed prior to publication. This study uses advanced technology but the data obtained appeared not explored enough, even the spatial transcriptomics is definitely an interesting analysis but in the end it appears to not add much at the meaning of manuscript. On their own, these data add little to the overall understanding of dopaminergic neuron development. It would be far more impactful if the authors explored the developmental layering of neurons and the factors driving the spatial segregation of dopaminergic subtypes. Overall, this work has significant potential, but it seems that the valuable resource the authors have created has not been fully leveraged:

Author response:

We thank the reviewer for their thoughtful and detailed feedback on our manuscript. We appreciate their recognition of the strengths of our study and the potential impact of our findings on the field of dopaminergic system development. We are also pleased to hear that they find our integration of single-cell datasets and the profiling of fetal tissue from 6-22 post-conceptional weeks to be a valuable and novel resource for the scientific community. We also appreciate their acknowledgment of the intriguing findings related to DTDS and the potential of our resource to analyze developmental trajectories, maturation, and the diversity of midbrain organoids.

We appreciate the reviewer recognizing the advanced technology used in this paper, in particular the use of spatial transcriptomics. We believe that the spatial transcriptomics analysis is integral to the underlying message of our paper since it allows the in-depth comparison between fetal tissues and 3D tissue-like organoids. Through these comparisons, we could understand cell-cell communication pathways and cell maturation behavior in a spatial context. Ultimately, this allowed us to advance the comparison at the tissue level, beyond single-cell analysis. This allowed a more rigorous, spatially informed quantification of similarity between midbrain tissue and in vitro tissue-like structures for the first time. Ultimately, this led to a novel temporal alignment between these samples whilst accounting for spatial characteristics. We have revised the text to better highlight the results yielded by spatial transcriptomics.

In response to reviewer 2's concerns regarding the exploration of our data, we have made several revisions to enhance the depth and impact of our analysis:

- We have conducted further exploratory analyses to uncover additional insights into dopaminergic neuron development, particularly focusing on trajectory analysis based on pseudotime. Additionally, we have examined expression patterns, signaling pathways, and cellular interactions that contribute to the differentiation and maturation of dopaminergic neurons.

- We have added analysis of patterning factors involved in midbrain specification across the timepoints of our study to our existing maturation trajectory analysis and identification of differentially expressed genes involved in maturation (Fig 5G). We have investigated cell-type-specific subclusters of dopaminergic neurons (PMID: 34911939) and investigated the spatial segregation of these dopaminergic subtypes.
- We have examined the spatial distribution of genes involved in maturation of the neural tube (PMID: 37824650) across all tissue timepoints, to better highlight the layering of developmental pathways (Fig 5).
- We have provided new insights into the genes associated with spatial maturation of dopaminergic neuron subtypes by annotating regions containing TH-expressing neurons in different spatial locations across the maturation path of dopaminergic neurons through the primordium of the SN and VTA at 17 PCW. This allowed us to identify differentially expressed genes across dopaminergic neurons based on spatial localization (Fig 6, p16-18, lines 476-512).
- We have improved the presentation of our data to ensure clarity and readability. This includes revising figures, tables, and supplementary materials to better convey our findings and their implications.

Major points:

1. The histological analysis of embryos 13, 15, 19 and 22 PCW is an important resource but needs extension to be fully informative. For example, expression of A9 vs A10 specific proteins ALDHA1A, PITX3, OTX2, GIRK2, CALB in the TH⁺ neurons should be analysed.

Author response:

We agree with the reviewer's comment and provide the requested analysis (Supplementary Figures 2-4 and page 5-6, lines 123-154).

2. Profiling a total of 34,191 cells from six donors, pooled into 13 samples (with 10X libraries prepared from a single channel), represents a relatively low number of profiled cells for a study aiming to create an in vivo and in vitro spatiotemporal atlas of human midbrain development. While we understand the challenges of working with precious fetal material, particularly from the second trimester—a real novel data source in this study—the limited number of dopaminergic neurons captured at single cell resolution emerging from these datasets limits the impact of the findings. Increasing the number of profiled cells (in particular DA neurons) would be necessary for robustness and overall impact of the findings. Maybe this can be achieved via nuclei seq?

Author response:

- We thank the reviewer for their comment and acknowledge that the number of cells presented here is limited for a single-cell atlas. Still, we believe that the integration of

complementary approaches (immunohistochemistry, single cell and spatial transcriptomics) provides valuable context and makes this a useful resource for the midbrain research community.

- Although increasing the number of profiled cells could be beneficial, we still capture the complexity of the developing human midbrain and we could still explore the degree of alignment between the 2D and 3D-derived models, as well with other single cell datasets. Furthermore, our single-cell resource provides new insights into the in vitro model of the neurological disorder Dopamine Transporter Deficiency Syndrome (DTDS). Overall, we have refined the focus of the manuscript and adjusted the title to better reflect the strengths and limitations of our data, rather than presenting it as a mere spatiotemporal atlas.
- Although single nuclei seq better captures mDA neurons, it cannot profile the cytoplasmic transcriptome of cells. In that scenario, the use of snRNA-seq would not have allowed us to track changes in mitochondrial gene expression, which are particularly relevant when comparing in vitro models or when monitoring diseases, e.g. DTDS progression. Also, the spatial cell type deconvolution could have been compromised given the reduced resolution in capturing the full transcriptome.

3. The presentation of different clusters (i.e. Fig 3B-D) gets very hard to digest with this many cell types. Especially 3C is almost impossible to decode. Would the clarity be improved by splitting the cells into major cell types (neurons, glia, progenitors, other) and then perform characterization/visualization per major cell type?

Author response:

We agree with the reviewer that Fig. 3C-D were difficult to interpret due to the large number of cell subtypes. To improve clarity, we now provide two levels of annotation:

- A **high-resolution** annotation (24 subtypes, as originally shown).
- A **low-resolution** annotation (10 clusters), informed by lower-resolution clustering (res=0.1) and grouped by developmental and lineage relevance.

The low-resolution annotation is listed in major point 12 from Reviewer 1. To enhance readability, Fig. 3 C-D have been updated to display only the low-resolution annotation. The high-resolution annotation is still present in the manuscript and can be visualized only in Supp Fig. 10B (faceted UMAP), without clarity being compromised. We believe keeping the high-resolution annotation is still important to capture differences in composition and pseudotime. We think this tiered approach offers both clarity and finer granularity when needed.

4. The developmental trajectories of dopamine (DA) neurons and non-neuronal cell types in the fetal samples should be traced so that the analysis not merely complement existing human datasets on midbrain development (as mentioned in line 512) but stands as a valuable reference in its own right.

Author response:

We thank the reviewer for this insightful suggestion. To ensure that our dataset represents a valuable reference for midbrain development, we performed additional trajectory analysis (page 10-11, Results section, lines 283-310) as follows:

i) Inference of trajectories (neuronal and astrocytic lineages):

We inferred developmental trajectories for neurons and astrocytes, following a modified approach from Caporale et al. 2025, *Nat. Methods* (PMID: 39653820). Briefly, we clustered the cells using the Leiden algorithm at high-resolution ($res=1$) and constructed a PAGA graph, pruning out edges with low weights ($w<0.05$). We then computed the shortest path from early midbrain progenitors to either dopaminergic neurons or astrocytes (Fig 3I, Supp Fig 11A-C, D-E, J-K).

ii) Exploring the changes along the developmental trajectory:

- Using only cells along the inferred trajectories, we found that the first principal component (PC1) captures the emergence of cell type identities across the neuronal and astrocytic lineages (Supp Fig 11F-H, Supp Fig 9L-M).

- We explored PC1 loadings to identify genes associated with progenitor-like states, differentiation or maturation, revealing known and novel markers (Fig 3J, Supp Fig 11I, 9O).

- For the astrocyte trajectory, we observed a very good correlation of each developmental time point (PCW10, 12, 16, 20) with the inferred pseudotime of cells (Supp Fig 11N).

- Finally, we modeled gene expression dynamics along both trajectories using tradeSeq (Van den Berge, K. et al. 2020, *Nat. Comms*, PMID: 32139671). We characterized the gene expression profiles of key markers, including morphogens, transcription factors, enzymes, transporters and structural genes, among others (Fig 3K, Supp Fig 11P). Notably, the expression of canonical dopaminergic markers, such as *GIRK2* and *TH*, peaked at mid-trajectory, with *TH* following similar expression dynamics along the maturation path inferred by spatial transcriptomics at 17 PCW (Fig 5E, Fig 6).

Further details about this trajectory analysis on the scRNA-seq data are provided in the main text and in the Methods section. Also, a new section in the discussion was added, to reflect this new analysis (Discussion section, page 22-23, lines 647-674).

5. The authors state that “generally, neuron proportions decreased at later stages of fetal development, with a simultaneous increase in glial cells, microglia, and oligodendrocyte cell lineages. Unsurprisingly, the proportion of neurons in adult post-mortem samples was only 3.3%.” While this is understandable, given that fetal tissue contains a greater variety of cell types compared to in vitro hPSC cultures, the authors should consider the possibility that tissue dissociation during single-cell preparation may disproportionately affect neurons compared to other cell types. Additionally, Figure 3D clearly shows that samples from PCW16 and PCW20—

representing the real novel data in this manuscript—contain very few DA neurons, which warrants further consideration. Using scRNA-seq rather than nuclei makes one wonder which fragile cell types (i.e. neurons) are missed?

Author response:

We thank the reviewer for raising this important point regarding potential dissociation biases during single-cell preparation, particularly the underrepresentation of fragile neuronal populations, including DA neurons, at later developmental stages. This phenomenon is also seen in previous published samples from the second trimester, as in Braun et al. 2013 (PMID: 37824650).

We agree that tissue dissociation can disproportionately affect fragile and highly arborized neurons, potentially leading to underrepresentation of these populations in single-cell suspensions. Despite this limitation, we believe our integration of *in vitro* and *in vivo* datasets enables the alignment of DA lineage progression and maturation, providing biologically meaningful insights into human midbrain development, even with the relatively low absolute DA neuron counts.

We agree that snRNA-seq could reduce dissociation bias and potentially improve recovery of fragile neuronal populations. However, we selected scRNA-seq for this study to enable comprehensive capture of cytoplasmic transcripts and to track mitochondrial gene expression, of key relevance in the context of DTDS progression. Additionally, we prioritized scRNA-seq to allow direct comparability with other recent human fetal midbrain datasets (e.g., La Manno et al., 2016; Fiorenzano et al., 2021; Braun et al., 2023; Zagare et al., 2022), all of which employed scRNA-seq rather than snRNA. In contrast, using snRNA-seq could have introduced an additional batch effect, complicating cross-dataset comparisons. Importantly, while dissociation bias may contribute to lower neuronal count, the observed decrease in neuronal proportions along with an increase in glial and oligodendrocyte lineage cells across developmental stages aligns with prior single-cell and bulk transcriptomic studies of human brain development, supporting the biological relevance of these trends.

We have revised the manuscript to explicitly acknowledge the potential underrepresentation of fragile neuronal subtypes, including DA neurons, due to dissociation limitations in our current approach while clarifying the rationale for using scRNA-seq to enable direct dataset comparisons (**Discussion section, lines 675-685**). We believe these clarifications strengthen the transparency and interpretability of our findings while highlighting the value of this unique dataset for advancing our understanding of human midbrain development.

6. The spatial transcriptomics used in this study has a low resolution, which limits its ability to provide detailed biological insights. As shown in Figure 5A–E, the resolution (~55 μm per spot) results in broad clustering of cell types rather than precise localization. For example, the spatial distribution of progenitor and post-mitotic cells lacks the granularity needed to separate closely packed cell populations or specific niches. While this method gives a general overview of cellular organization, it should be complemented with supporting immunostainings.

Author response:

While we appreciate that the Visium platform we used here has limited resolution, the sequencing data it provides has allowed us to carry out many analyses that would have been impossible with the image-based, higher-resolution technologies available at the time of generation of this dataset. Namely, the sequencing data from the Visium platform enabled:

- identification of differentially expressed genes across maturation trajectories;
- identification of novel ligand-receptor pairs;
- rigorous quantification of spatial similarity.

We also note that we carried out cell type deconvolution (based on our single cell transcriptomics of the same regions) to estimate the proportion of cell types at each 55µm spot, which mitigates the limited resolution to some extent. In any case, we acknowledge the reviewer's comment about the 55 µm resolution of the Visium platform. For this reason, we incorporated immunofluorescence analysis of the dopaminergic neuron and other neural subtypes at the same post-conception weeks (Fig 6, Fig 7, Supp Fig 14).

Minor points:

7. The exact number of cells per replicate needs to be clearly stated in the main text or figures.

Author response:

We now provide the exact number of profiled cells per donor in the main text and an overview of our dataset in Sup. Fig 10A-B. Additionally, we incorporate a detailed explanation in Methods, as follows:

“For fetal samples, four 10x libraries were generated; while for the in vitro models (2D and 3D), nine libraries were generated - three per differentiation timepoint (D40, D70 and D120). In both cases, donors were genotyped prior to pooling for single-cell library preparation.

For fetal samples, donors from either the first trimester (PCW10, PCW12) or second trimester (PCW16, PCW20) were pooled together, and two technical replicates were generated per pool.

For in vitro samples, we randomized the pool composition to include differentiations from four donors at a time, either modelled in 2D or 3D. We avoided mixing samples with the same genetic background, either generated from different models (i.e. Control1-2D with Control1-3D), or corrected from a patient-derived line (i.e. Crispr1-3D with Patient2-3D). This strategy enables unambiguous deconvolution of donor identity later on.

Each donor/model combination is represented in at least two 10x libraries (range: 2-9 replicates). When combining replicates, the total number of cells per donor/model combination ranges between 1,346 and 9,394.”

8. Why not use the established acronym VLMC instead of “vascLepto”?

Author response:

We thank the reviewer for this observation and have accordingly replaced “vascLepto” with the established acronym “VLMC” across text and figures.

9. 6D, legend?

Author response:

We originally did not include a legend here since the figure 7D (old figure 6D) is simply a schematic demonstrating how the quantification is carried out. The important aspect of the figure is to show that the number of color-color neighboring spots are counted for each sample: the identities of the cell types shown in the schematic are irrelevant. Nevertheless, to avoid confusion to the reader, we now include a new figure legend that reads as follows: “Schematic of similarity quantification approach: cell types of neighboring spots are counted and used to compute similarity between samples.”

10. In fig 7, does (C) refer to d40 and (D) to d70?

Author response:

Yes, quantification in C refers to d40 controls (Control 1, Control 2, and CRISPR) and patient (Patient 1 and Patient 2) derived midbrain organoids, while quantification in D refers to d70. We have included this information in the figure and in the figure legend.

11. For the data in Fig 7, where there any cell type abundance differences between patients and controls?

Author response:

We thank the reviewer for requesting this relevant comparison between patient- and control-derived 3D models of DTDS syndrome. Previously, we observed by immunofluorescence a reduction in the number of TH⁺ and TH⁺/MAP2⁺ neurons in patient-derived lines from day 70 onwards (new Fig 8B-D), but we had not directly addressed this question using our single-cell dataset.

In response to this point, we performed new differential abundance analyses using miloR (Supplementary Review Fig 1). These tests revealed no significant differences in 3D organoids composition between patient and control-derived lines, either when adjusting for time (~Time + Condition), or when testing within individual timepoints (~Condition). These results (page 20) were consistent across both high- and low-resolution annotations. That said, by day 120 we observed broad shifts in composition (log₂FC) that, while not reaching statistical significance, suggest a decrease in specific dopaminergic neurons subtypes (hDA2, hNbDA) or more broadly, a decrease in neurons and immature neurons. Those observations align with the reduction in mDA seen by immunofluorescence.

Supplementary Review Fig.1. Differential abundance analysis of cell type composition (miRo) between patient- and control-derived 3D-models of Dopamine Transporter Deficiency Syndrome (DTDS). A-B, Tests using high and low-resolution annotations, respectively, adjusting for developmental timepoint as a covariate. **C-D,** Tests using high-resolution annotation for days 40 and 70. **E-F,** Tests using high and low-resolution annotations, respectively, at day 120. In both cases, we observe a reduction in specific dopaminergic neurons subtypes and glial cell progenitors (in blue), although these differences do not reach statistical significance.

Reviewer #4

Remarks to the Author:

Author response: We thank Reviewer 4 for co-reviewing our manuscript.

Remarks on code availability:

Yes, I have assessed the code, and I find it to be a well-prepared and valuable resource for the community. The results of the paper are reproducible to a significant extent, as the code is clearly structured and provides all the necessary components to replicate the experiments described.

Author response: We thank Reviewer 4 for their analysis of our codes and for finding it reproducible, well structured, and a valuable resource for the community.

Reviewer #6

Remarks to the Author:

Author response: We thank Reviewer 6 for co-reviewing our manuscript.